# The WHHERE coactivator complex is required for retinoic acid-dependent regulation of embryonic symmetry

Gonçalo C. Vilhais-Neto[1,2], Marjorie Fournier[1], Jean-Luc Plassat[1], Mihaela E. Sardiu[2], Anita Saraf[2], Jean-Marie Garnier[1], Mitsuji Maruhashi[1,2], Laurence Florens[2], Michael P. Washburn [2,3] & Olivier Pourquié [1,2,4,5,6]

Bilateral symmetry is a striking feature of the vertebrate body plan organization. Vertebral precursors, called somites, provide one of the best illustrations of embryonic symmetry. Maintenance of somitogenesis symmetry requires retinoic acid (RA) and its coactivator Rere/Atrophin2. Here, using a proteomic approach we identify a protein complex, containing Wdr5, Hdac1, Hdac2 and Rere (named WHHERE), which regulates RA signaling and controls embryonic symmetry. We demonstrate that Wdr5, Hdac1, and Hdac2 are required for RA signaling in vitro and in vivo. Mouse mutants for *Wdr5* and *Hdac1* exhibit asymmetrical somite formation characteristic of RA-deficiency. We also identify the Rere-binding histone methyltransferase Ehmt2/G9a, as a RA coactivator controlling somite symmetry. Upon RA treatment, WHHERE and Ehmt2 become enriched at RA target genes to promote RNA polymerase II recruitment. Our work identifies a protein complex linking key epigenetic regulators acting in the molecular control of embryonic bilateral symmetry.

---

[1] Institut de Génétique et de Biologie Moléculaire et Cellulaire (IGBMC), CNRS (UMR 7104), Inserm U964, Université de Strasbourg, Illkirch F-67400, France. [2] Stowers Institute for Medical Research, Kansas City, MO 64110, USA. [3] Department of Pathology and Laboratory Medicine, University of Kansas Medical Center, Kansas City, KS 66160, USA. [4] Department of Anatomy and Cell Biology, University of Kansas Medical Center, Kansas City, KS 66160, USA. [5] Howard Hughes Medical Institute, Kansas City, MO 64110, USA. [6] Department of Genetics, Harvard Medical School and Department of Pathology, Brigham and Women's Hospital, 60 Fenwood Road, Boston, MA 02115, USA. Correspondence and requests for materials should be addressed to O.P. (email: pourquie@genetics.med.harvard.edu)

The development of bilaterally symmetrical structures, such as limbs or somites takes place concomitantly with the asymmetric formation of internal organs such as heart, gut, and liver. Whereas the pathway responsible for establishing left–right identity in the embryo begins to be well understood[1], little is known about the mechanisms controlling embryonic symmetry. Retinoic acid (RA) is a derivative of vitamin A, signaling via a heterodimeric RAR/RXR nuclear receptor transcription factor[2–4]. In the absence of RA, the heterodimer binds target genes together with the SMRT and NCoR corepressor complexes and histone deacetylases such as Hdac3 to silence gene expression. When the RA ligand binds to RAR, the corepressors are replaced by a set of coactivators including histone acetyltransferases, contributing to active transcription of RA target genes[5, 6]. In the absence of RA signaling in the mouse embryo, somite formation becomes asymmetrical, showing a significant delay on the right side[7]. A similar somite desynchronization phenotype is also observed in mutants for the protein Rere (or Atrophin2) which acts as a coactivator for RA signalling[8].

Here, we identify and characterize biochemically a previously undescribed retinoic coactivator complex, containing the proteins Wdr5, Hdac1, Hdac2, and Rere (the WHHERE complex). We demonstrate that mouse mutants of the WHHERE complex members down-regulate RA signaling and exhibit somite bilateral symmetry defects. We report that the WHHERE complex also binds the histone methyltransferase Ehmt2/G9a. Null mouse mutants for *Ehmt2* also exhibit RA downregulation and somite symmetry defects indicating that it also acts as an activator of RA signaling. We show that Ehmt2 and the WHHERE complex bind to the promoter of RA targets and serve to recruit PolII to trigger gene activation.

## Results

### Identification of the WHHERE complex.
In order to understand the mechanism of action of Rere in the RA pathway controlling somite symmetry, we first set out to identify Rere-interacting proteins in the mouse mesoderm. To that end, we generated a transgenic mouse line, which allows the conditional expression of a tagged version of Rere containing two HA epitopes at the C-terminal end of the protein (Rere-HA). A *Rere-HA* construct preceded by a *LoxP-STOP-LoxP* cassette was introduced into the *Rosa26* locus by homologous recombination in mouse embryonic stem (ES) cells. We then used these cells to generate a *Rosa26-LoxP-STOP-LoxP-Rere-HA* mouse line (*RS-Rere-HA* line). Whereas *Rere* mutants (*Rere^om/om^*) die around E9.5 with defects in forebrain, heart and a right side specific delay in somite formation[8, 9], expression of at least one *Rere-HA* allele in the mutant *Rere*^om/om^ background led to morphologically normal embryos (Supplementary Fig. 1a–c). Therefore, the tagged Rere-HA protein is functional in vivo.

To direct expression of Rere-HA to the mesoderm, *RS-Rere-HA* mice were crossed to the *T-Cre* mouse line[10]. We prepared whole cell protein extracts from ~600 *RS-Rere-HA;T-Cre* embryos, and performed affinity purification using anti-HA antibodies under high or low salt conditions (Fig. 1a). To identify the immunoprecipitated proteins, eluted fractions were submitted to mass spectrometry analysis using the multidimensional protein identification technology (MudPIT)[11]. A set of 105 common proteins was found between the different immunopurification conditions (Supplementary Figs. 1d, 6 and Supplementary Data 1). Hierarchical clustering analysis of the 105 proteins based on the normalized spectral abundance factor (NSAF)[12] in the different immunoprecipitation conditions identified three abundant proteins tightly clustering with Rere (Supplementary Fig. 1e, f). These include Rere's known binding partners Hdac1

and Hdac2 as well as a previously unreported partner, Wdr5[13–16] (Fig. 1b, Supplementary Figs. 2a and 9). To estimate relative protein levels, we compared the NSAF values for each protein[12]. NSAF for Rere, Hdac1, and Hdac2 were similar while it was three times higher for Wdr5 (Fig. 1b). These results suggest that the proteins Rere, Hdac1, Hdac2, and Wdr5 can interact in the mesoderm.

### In vitro characterization of the WHHERE complex.
To validate the identification of Wdr5 as an interacting partner of Rere, Hdac1, and Hdac2, we co-expressed tagged versions of the four proteins (Rere-Flag, Flag-Hdac1, Flag-Hdac2, and HA-Wdr5) using a baculovirus-insect cell expression system. After Flag immunoprecipitation of Rere, Hdac1 and Hdac2, HA-Wdr5 was detected in the eluates, as confirmed by LC–MS/MS analysis of the Coomassie stained gel bands (Fig. 1c, Supplementary Figs. 2b and 7). The four proteins still co-purified together at high salt concentration (500 mM KCl) suggesting the existence of a stable protein complex comprising Rere, Hdac1, Hdac2, and Wdr5 (Fig. 1c and Supplementary Fig. 7). By co-immunoprecipitation of baculovirus-expressed Rere-Flag and HA-Wdr5, we could demonstrate that Rere binds directly to Wdr5 (Fig. 1d and Supplementary Fig. 7). Wdr5 does not bind directly to Hdac1 or Hdac2 in high or low salt wash conditions (Supplementary Figs. 2c and 9). Wdr5 binding to Hdac1 and Hdac2 could only be detected when Rere is also co-expressed (Supplementary Figs. 2c and 9), suggesting that Rere acts as a scaffolding component binding Hdac1/Hdac2 and Wdr5. To analyze whether the four co-immunopurified proteins form a stable protein complex, we carried out gel filtration chromatography followed by western blot and mass spectrometry analysis. All four proteins co-eluted together in a fraction corresponding to a high molecular weight complex of 0.5–0.6 MDa (Figs. 1e, f, Supplementary Figs. 2d, e and 7). This molecular weight is consistent with the abundance predicted by the NSAF values of the proteomic analysis. Additionally, Hdac1, Hdac2, and Wdr5 co-immunoprecipitated with the endogenous Rere in NIH3T3 cells further supporting the existence of such a protein complex (Fig. 1g and Supplementary Fig. 7). Altogether, these results suggest that Wdr5, Hdac1, Hdac2 and Rere can form a stable protein complex, which we called the WHHERE complex.

### WHHERE complex members act as coactivators of RA signaling.
To test the coactivator properties of WHHERE on RA signaling, we transfected NIH3T3 fibroblast cells with expression vectors coding for either *Rere, Wdr5, Hdac1,* or *Hdac2*, together with a RA reporter containing the well-characterized retinoic acid response element (RARE) of the known RA target *retinoic Acid Receptor beta* (*Rarβ*)[17], driving luciferase expression. In the presence of RA, overexpression of each of the four WHHERE complex proteins increased RA reporter expression (Fig. 2a). This activation was not observed in the absence of RA (Supplementary Fig. 3a). Also overexpression of the WHHERE complex components did not activate either a minimal or a SV40 promoter driving luciferase supporting the specificity of the RA response (Supplementary Fig. 3b, c). Furthermore, co-transfection of *Hdac1* and *Hdac2* cooperated to activate RA signaling (Fig. 2b). Conversely, treatment of fibroblast cultures with siRNAs against *Rere, Hdac1,* or *Wdr5* in the presence of RA decreased RARE-Luciferase reporter activity (Fig. 2c and Supplementary Fig. 3d–f). Treatment with *Hdac2* siRNA led to an increase of RARE-Luciferase reporter activity (Fig. 2c and Supplementary Fig. 3g), which might be explained by a stabilization of Hdac1 (due to the decrease of Hdac2), potentially resulting in an increase in RA

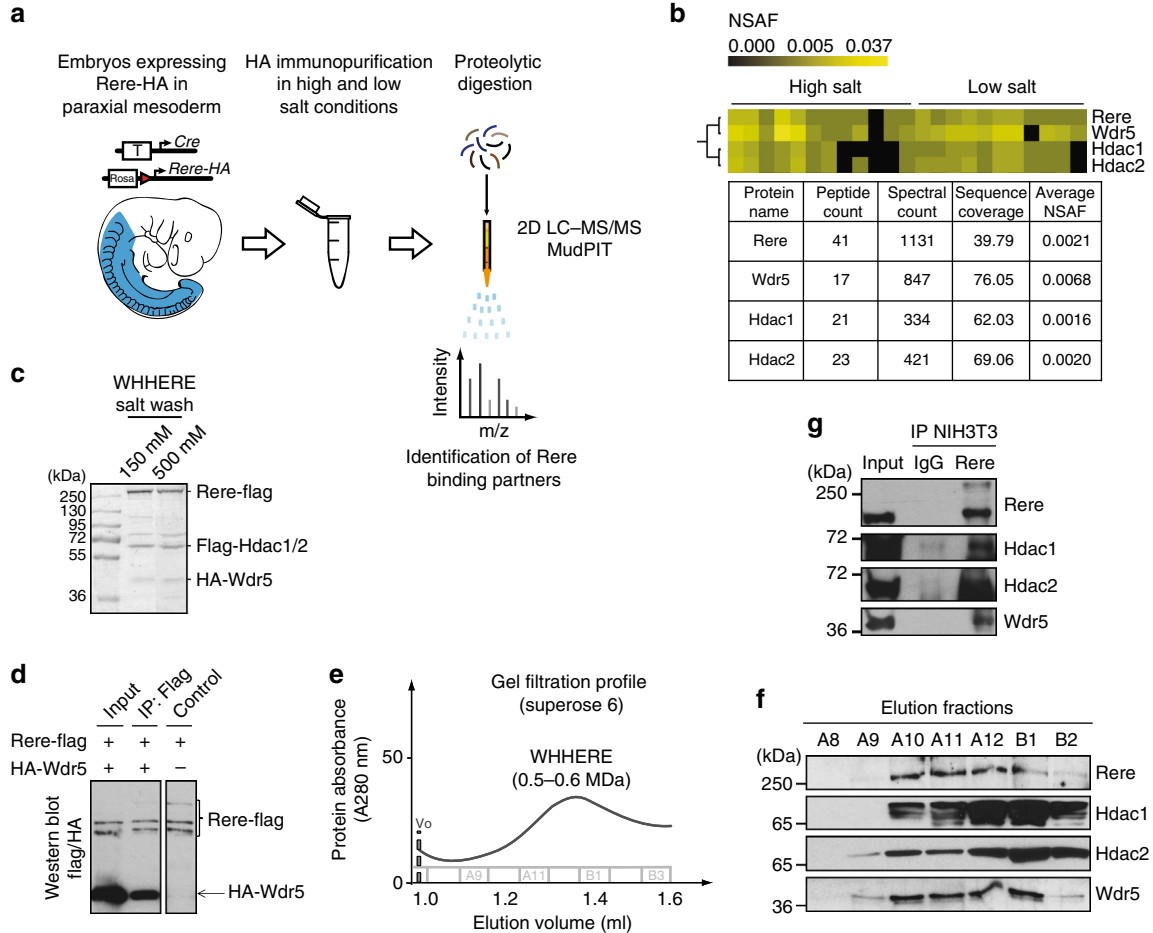

**Fig. 1** Proteomic identification of the WHHERE protein complex containing Wdr5, Hdac1, Hdac2, and Rere. **a** Schematic workflow for the proteomic strategy used to identify Rere-associated proteins. **b** Hierarchical clustering of the NSAF values showing Wdr5, Hdac1, and Hdac2 clustering with Rere identified by MudPIT. *Columns* correspond to different immunoprecipitations conditions. Table showing the peptide count, spectral count, sequence coverage (%) and average NSAF values for Rere, Wdr5, Hdac1, and Hdac2. **c** SDS-PAGE gel stained with Coomassie blue after Flag immunopurification at low (150 mM KCl) and high (500 mM KCl) salt wash of the recombinant WHHERE complex from baculoviruses infected cells co-expressing Rere-Flag, Flag-Hdac1, Flag-Hdac2, and HA-Wdr5. Identification of the different components was confirmed by LC–MS/MS analysis of the corresponding gel bands. **d** Flag immunoprecipitation from extracts of cells infected with baculoviruses expressing HA-Wdr5 and Rere-Flag. Flag and HA western blot. The three bands recognized by the anti-Flag antibodies correspond to Rere and its degradation products. **e** Gel filtration chromatography profile of the WHHERE complex purified from baculoviruses infected cells co-expressing Rere-Flag, Flag-Hdac1, Flag-Hdac2, and HA-Wdr5. **f** Immunoblot analysis for Rere, Hdac1, Hdac2, and Wdr5 showing the gel filtration chromatography elution fractions from A8 to B2. **g** Rere immunoprecipitation from NIH3T3 cells identifying the endogenous WHHERE complex components. Rere, Hdac1, Hdac2, and Wdr5 were detected after western blot analysis of Rere immunoprecipitation eluates and compared to a negative control immunoprecipitation (IgG) and input extract

signalling[18, 19]. Consistent with this possibility, the double knockdown of *Hdac1* and *Hdac2* further decreased RA signaling compared to *Hdac1* siRNA alone (Fig. 2d). Furthermore, *Hdac1* or *Hdac1/Hdac2* depletion reduced RA activation mediated by *Rere* and *Wdr5* (Fig. 2e, f). Inhibition of deacetylase enzymatic activity with a range of chemical inhibitors decreased RA signaling (Fig. 2g)[20]. In line with this, overexpression of *Rere* or *Wdr5* did not lead to a significant increase in RA signaling when cells were treated with the HDAC inhibitors Trichostatin A (TSA) or sodium butyrate (SB) (Fig. 2h, i). Hdac1 and Hdac2 have been shown to bind Rere N-terminal region[9, 13, 14]. In the presence of RA, overexpression of the N-terminal region of Rere (N-Rere) strongly increased RA signaling whereas no activation could be seen with Rere C-terminal region (Rere C) (Fig. 2j). The activation by N-Rere was dependent on deacetylase activity since TSA or SB treatment strongly decreased N-Rere-dependent RA signaling (Fig. 2k). Overall, these results show that the WHHERE complex proteins Rere, Wdr5, Hdac1, and Hdac2 act to activate

the RA pathway in NIH3T3 cells. Moreover, activation of RA signaling by the WHHERE complex depends on Hdac1 and Hdac2 deacetylase activity.

**WHHERE regulates RA signaling and somite bilateral symmetry.** To analyze WHHERE-dependent RA regulation in vivo, we characterized the phenotype of mouse embryos mutant for the different components of the complex. *Hdac1* mutants exhibit a variety of developmental defects similar to *Rere*[om/om] embryos[21]. We introduced the *RARE-LacZ* reporter[22] in *Hdac1*-null embryos and observed a strong downregulation of LacZ expression, similar to that seen in *Rere* mutants (Fig. 3a–c). *Hdac2* mutants do not show developmental defects and can survive until the perinatal period[21]. To analyze Wdr5 function in vivo we first set out to generate a conditional knockout mouse line by introducing *LoxP* sites flanking exons 2 to exon 4 (Supplementary Fig. 4a). No heterozygous embryos were recovered at E8.5 after crossing

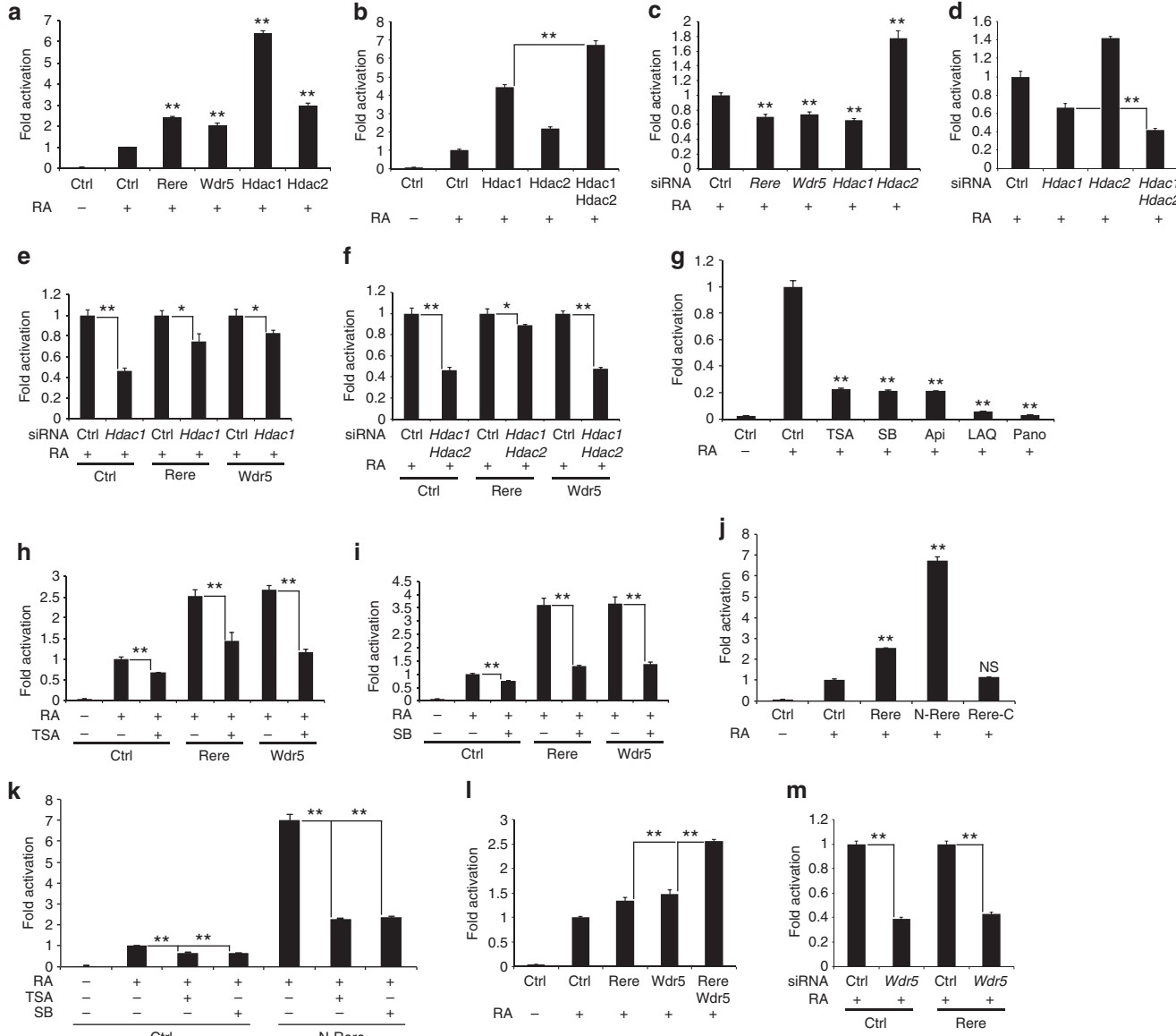

**Fig. 2** The WHHERE complex acts as a coactivator for retinoic acid signaling. **a-k** RARE-Luciferase activity from NIH3T3 cells treated or not with 1 µM RA for 20 h. **a** Cells transfected with expression plasmids containing *Rere*, *Wdr5*, *Hdac1*, or *Hdac2* ($n = 4$). **b** Cells transfected with expression plasmids containing *Hdac1*, *Hdac2*, or both ($n = 4$). **c** Cells treated either with siRNA for *Rere*, *Wdr5*, *Hdac1*, or *Hdac2* ($n = 4$). **d** Cells treated with siRNA for *Hdac1*, *Hdac2* or both ($n = 4$). **e** Cells overexpressing *Rere* or *Wdr5* and treated with siRNA against *Hdac1* ($n = 4$). **f** Cells overexpressing *Rere* or *Wdr5* and treated with siRNA against both *Hdac1* and *Hdac2* ($n = 4$). **g** Cells treated with the HDAC inhibitors Trichostatin A (TSA) (60 nM), sodium butyrate (SB) (3 mM), apicidin (Api) (300 nM), LAQ824 (LAQ) (60 nM) and Panobinostat (Pano) (30 nM) ($n = 4$). **h** Cells overexpressing *Rere* or *Wdr5* and treated with TSA (30 nM) ($n = 4$). **i** Cells overexpressing *Rere* or *Wdr5* and treated with SB (1.5 mM) ($n = 4$). **j** Cells transfected with expression plasmids containing *Rere*, *N-Rere* (*Rere* N-terminal domain) or *Rere C* (*Rere* C-terminal domain) ($n = 3$). **k** Cells overexpressing *N-Rere* (*Rere* N-terminal domain) and treated with TSA (30 nM) or SB (1.5 mM) ($n = 4$). **l** Cells transfected with expression plasmids containing *Rere*, *Wdr5*, or both ($n = 4$). **m** Cells overexpressing *Rere* and treated with siRNA for *Wdr5* ($n = 4$). In all *graphs* data represent mean ± s.e.m. NS—not significant, *$P < 0.05$ and **$P < 0.01$. Student's unpaired two-tailed *t*-test

with an ubiquitous Cre suggesting that the *Wdr5* mutation is heterozygous lethal at early stages. To circumvent this issue, we crossed mice homozygous for the *Wdr5* conditional allele (*Wdr5^{fl/fl}*) to the mesoderm-specific *T-Cre* line[10]. Removal of a single *Wdr5* allele in *Wdr5^{fl/+};T-Cre;RARE-LacZ* embryos was sufficient to strongly decrease β-galactosidase staining in the mesoderm (Fig. 3d). In contrast, no significant downregulation of Notch target genes such as *Lfng* or *Hes7* was detected in these mutants suggesting the Notch-dependent somite segmentation clock appears normal (Supplementary Fig. 4b–e). Altogether, these results support the function of the WHHERE complex in the control of RA signaling in the embryo.

We next examined the effect of null mutations of members of the WHHERE complex on somite symmetry. In nearly half of the *Hdac1^{−/−}* embryos, we observed a right side delay of somite formation resembling the defect observed in *Rere^{om/om}* and *Raldh2^{−/−}* mutants[7, 8] (Fig. 3e–g, i, j and Supplementary Fig. 4f). Similar lateralized somite desynchronization defects were observed in a limited number of *Wdr5^{fl/+};T-Cre* embryos which also exhibited broad somitogenesis defects (Fig. 3h). We next intercrossed *Wdr5^{fl/+}* and *Rere;T-Cre* mice. Strikingly *Rere^{om/om};Wdr5^{fl/+};T-Cre* embryos exhibited a stronger phenotype than either *Rere^{om/om}* or *Wdr5^{fl/+};T-Cre* (Fig. 3l–o) with ~ 80% (7 out of 9) of the embryos with a right side delay in

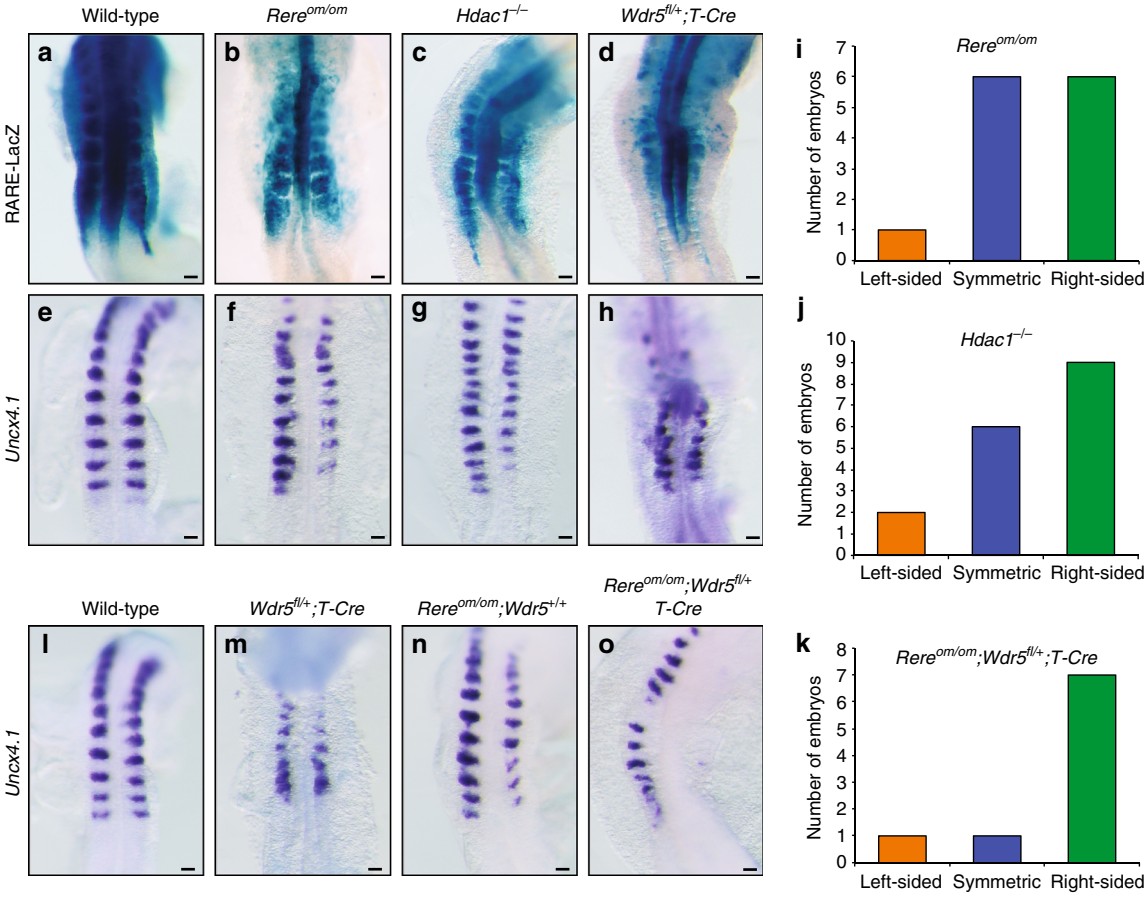

**Fig. 3** The WHHERE complex controls somite bilateral symmetry. **a–d** RARE-LacZ expression in wild-type **a**, *Rere*^*om/om*^ **b**, *Hdac1*^−/−^ **c** and *Wdr5*^*fl/+*^; *T-Cre* **d** embryos at E8.75-E9.0 (*dorsal views*). **e–h** In situ hybridization showing somites labeled with *Uncx4.1* in wild-type **e**, *Rere*^*om/om*^ **f**, *Hdac1*^−/−^ **g** and *Wdr5*^*fl/+*^; *T-Cre* **h** embryos at E8.75-E9.0 (*dorsal views*). **i–k** Graphs representing the number of 7- somite to 15-somite stage embryos with left-sided (*orange*), symmetric (*blue*) or right-sided (*green*) delay in somite formation: *Rere*^*om/om*^ **i**, *Hdac1*^−/−^ **j** and *Rere*^*om/om*^;*Wdr5*^*fl/+*^;*T-Cre* **k**. **l–o** In situ hybridization for *Uncx4.1* in wild-type **l**, *Wdr5*^*fl/+*^;*T-Cre* **m**, *Rere*^*om/om*^;*Wdr5*^*+/+*^ **n** and *Rere*^*om/om*^;*Wdr5*^*fl/+*^;*T-Cre* **o** embryos at E8.75-E9.0 (*dorsal views*). For each genotype at least 5–10 embryos were analyzed. Bar = 100 microns

somitogenesis (Fig. 3k and Supplementary Fig. 4f) and in ~ 35% (3 out 9) of the embryos, no clear segmented structures forming on the right side (Fig. 3o and Supplementary Fig. 4g, h). This provides strong evidence for genetic interaction between *Wdr5* and *Rere* in the control of somite symmetry. Consistently, transfection of *Rere* and *Wdr5* together in NIH3T3 cells activated RA signaling more strongly than overexpression of either of the proteins alone (Fig. 2l) and *Rere*-dependent RA activation is no longer observed in cells depleted of *Wdr5* (Fig. 2m). Overall, these data reveal the importance of WHHERE in positive regulation of RA signaling and in the control of somitogenesis synchronization in mouse embryos.

**WHHERE is required to recruit PolII to RA targets' promoter.** Next we analyzed the binding of the WHHERE complex to RA regulated genes by chromatin immunoprecipitation (ChIP). To that end, we first checked the expression kinetics of the RARE-Luciferase reporter and of the endogenous *Rarβ* gene in NIH3T3 cells treated with RA for 1, 2, and 6 h. At 2 and 6 h, strong transcriptional activation was observed for both RA targets (Fig. 4a, b). In mouse embryos, we observed by ChIP analysis that the retinoic acid receptor alpha (Rarα) and all the WHHERE complex components were present at the *Rarβ* promoter and at the RARE-containing reporter *RARE-LacZ* (Fig. 4c–f). In ChIP experiments performed 1 h after RA treatment of NIH3T3 cells, Rere, Wdr5, Hdac1, and Hdac2 but not Rarα increased at

the *Rarβ* promoter (Fig. 4g, h). This effect was specific, as no significant enrichment of WHHERE complex members could be observed in regions upstream of the *Rarβ* promoter in similar conditions (Fig. 4h, *bottom graph*). Recruitment of RNA Polymerase II (Pol II) increased following RA treatment paralleling the WHHERE complex recruitment (Fig. 4i). Then we investigated the requirement of Hdac1 deacetylase activity in transcription activation and Pol II recruitment during RA signaling. Decreased RARE-Luciferase and *Rarβ* expression was observed in TSA-treated cells after RA treatment demonstrating the importance of HDAC activity in early activation of RA target genes (Fig. 4a, b). TSA treatment or siRNA-mediated knockdown of *Hdac1* decreased Pol II recruitment at the *Rarβ* promoter (Fig. 4j, k). In contrast, no change in Pol II occupancy levels could be detected in RA target genes unresponsive to RA in NIH3T3 cells, such as *Cyp26a1* and *Hoxa1* or in promoters of genes unresponsive to RA signaling following such treatments (Supplementary Fig. 3k–m). In transfected cells expressing Flag-Hdac1 alone or Flag-Hdac1 and Rere-HA, we observed binding of Hdac1 to endogenous Rarα. This suggests that Hdac1 could bridge the WHHERE complex to Rarα (Fig. 4l–n and Supplementary Fig. 8). Together, these data support a role of the WHHERE complex in the recruitment of Pol II necessary for early activation of RA regulated genes. This role depends on the deacetylase activity of Hdac1 and Hdac2.

**Ehmt2 acts together with WHHERE to activate RA signaling.** The histone methyltransferase Ehmt2 (G9a) was shown to bind the N-terminal SANT domain of Rere, and together with Hdac1/Hdac2 to regulate the methylation of H3K9 at specific loci leading to the formation of compact heterochromatin and gene silencing[14]. In the proteomic experiment, Ehmt2 (and the related protein Ehmt1) were detected with low NSAF values compared to

the members of the WHHERE complex suggesting that its binding to Rere might be transient (Supplementary Fig. 5a). In mouse embryos deficient for Ehmt2[23] crossed to the RARE-LacZ reporter[22], LacZ expression is downregulated suggesting that Ehmt2 is also implicated in positive regulation of RA signaling (Fig. 5a, b). Furthermore, half of the Ehmt2 mutant embryos presented a delay in somite formation on the right side

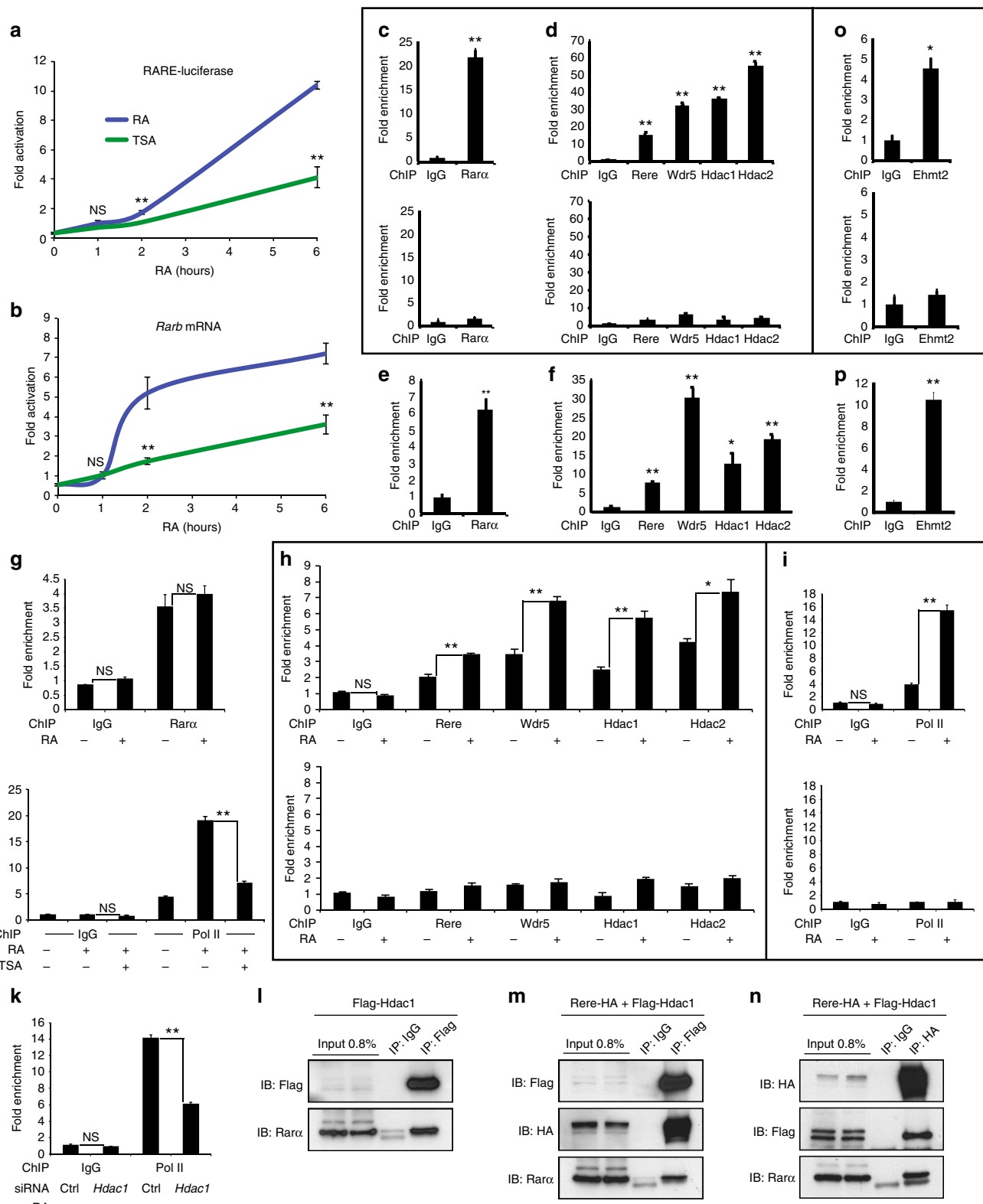

resembling mutants of members of the WHHERE complex (Fig. 5c–e; Supplementary Fig. 4f). In NIH3T3 cultures, siRNA-mediated knockdown of *Ehmt2* led to a downregulation of the RARE-Luciferase reporter activity (Fig. 5f and Supplementary Fig. 3h), whereas overexpressing Ehmt2 (or Ehmt1) stimulates RA signaling (Fig. 5g and Supplementary Fig. 5b). Co-transfection of *Ehmt2* with *Rere* increases the RA response more than transfection of either one alone (Fig. 5h). siRNA-mediated knockdown of *Ehmt2* inhibited Rere and Hdac1-dependent activation of the RA pathway (Fig. 5i). In NIH3T3 cells, Ehmt2 was recruited at the RARE element of the *Rarβ* promoter after 1 h of RA treatment while no such enrichment was observed in upstream regions (Fig. 5j and Supplementary Fig. 5c). In mouse embryos Ehmt2 could also be detected at the RARE element present in both the *Rarβ* promoter and the *RARE-LacZ* reporter (Fig. 4o, p). Knockdown of *Ehmt2* in NIH3T3 cells reduced the occupancy levels of Rere, Wdr5, Hdac1, Hdac2, and Pol II at the *Rarβ* promoter (Fig. 5k, l and Supplementary Fig. 5d). In NIH3T3 cells, inhibition of Ehmt2 methyltransferase activity with UNC0638 (U38) or UNC0646 (U46)[24, 25] did not alter RARE-Luciferase reporter activity or *Rarβ* mRNA expression (Supplementary Fig. 5e, f). Also no difference in H3K9me1 nor H3K9me2 levels was observed by ChIP after 1 h of RA treatment suggesting that Ehmt2 function is independent from its methyltransferase activity (Supplementary Fig. 5g, h). This suggests that Ehmt2 could function as a scaffold protein to stabilize the WHHERE complex at RA regulated genes to allow Pol II loading. Thus, these results indicate that Ehmt2 acts together with the WHHERE coactivator complex in the RA-dependent control of somite bilateral symmetry.

**Effect of RA signaling on key histone marks**. Wdr5 is part of complexes such as ATAC, MOF, and MLL, that are involved in histone acetylation and H3K4 methylation, which are chromatin modifications associated with transcriptional activation[26]. Despite the requirements of HDAC enzymatic activity for WHHERE function, we observed an increase of the levels of acetylated H3 and H4 as well as H3K27ac on the *Rarβ* promoter upon 1 h of RA treatment of NIH3T3 cells (Fig. 6a–c). This suggests that Hdac1 and Hdac2 might act on non-histone substrates or both could participate to the stability of the WHHERE complex. The H3K36me3 mark, which is associated to transcription elongation, was also increased (Fig. 6d). We also found that in the same conditions, H3K27me3 is absent from the *Rarβ* promoter (Fig. 6e), whereas H3K4me1 increases while H3K4me2 and H3K4me3 decrease (Fig. 6f–h). MLL3 and MLL4 are complexes which contain Wdr5 and regulate the deposition of the H3K4me1 mark[27]. The MLL3 and MLL4 complexes have been shown to be involved in RA-dependent transcription and Wdr5 might provide a link with the WHHERE complex[28]. Whether these complexes are also required for the WHHERE-dependent Pol II recruitment to the promoter of RA targets remains to be investigated.

**Negative regulation of RA signaling by Ep300**. The histone acetyl transferase Ep300 (p300) has been shown to acetylate Hdac1 leading to a decrease of its deacetylase activity[29]. In the presence of RA, overexpression of *Ep300* in NIH3T3 cells decreased RA signaling and inhibited Hdac1-dependent RA activation (Fig. 7a). In line with this, transfection of an *Hdac1* mutant form resistant to Ep300 acetylation (H1-6R)[29] activated more strongly the RA pathway than wild-type Hdac1 (H1-WT) (Fig. 7b). Moreover, Rarα- and Rere-dependent RA activation was inhibited following *Ep300* overexpression whereas transfection of the histone acetyl transferase *Kat2a* (*Gcn5*) together with *Rarα* or *Rere* increased RA signaling more than transfection of either construct alone (Fig. 7c, d). While treatment of fibroblast cultures with siRNAs against *Rere* or *Kat2a* in the presence of RA decreased RA reporter activity, knockdown of *Ep300* did not affect the RA pathway (Fig.7e and Supplementary Fig. 3i, j). Similarly in *Ep300* mutant embryos (*Ep300*[−/−])[30], the RARE-LacZ reporter[22] expression appeared normal (Fig. 7f, g) and somito-genesis progressed symmetrically (Figs. 7h, i). Altogether these results demonstrated that Ep300 negatively regulates Hdac1 activation of RA signaling. Furthermore Kat2a can participate together with the WHHERE complex in the activation of the RA pathway.

**Discussion**

The molecular mechanisms controlling embryonic bilateral symmetry are still poorly understood. In mouse, the only genes shown to act in this process are *Raldh2* and *Rere* which are involved in the control of RA signaling[7, 8]. Here, we identify a protein complex, called WHHERE which associates key epigenetic regulators such as Wdr5, Hdac1, and Hdac2 to the control of bilateral symmetry downstream of RA signaling. In the background of a situs inversus mutation (a mutation that can reverse left–right identity of the body), *Raldh2* and *Rere*-deficient mouse embryos can show a reversed somite defect, i.e., left-side delay of somitogenesis[8, 31]. This therefore argues that WHHERE-dependent RA signaling buffers a desynchronizing influence of the left–right determination pathway, involved in asymmetrical development of internal organs. RA signaling was shown to directly antagonize the left identity determinant *Fgf8* in the mouse embryo through recruitment of Hdac1 and Prc2, providing a potential explanation for this buffering effect[32, 33]. Thus, our work identifies a new pathway antagonizing FGF signaling acting downstream of RA to control bilateral symmetry in the mouse embryo.

Strikingly most of the members of the WHHERE complex including Rere, Hdac1, Hdac2, and also Ehmt2 have been

**Fig. 4** The WHHERE complex is recruited to the promoter of retinoic acid target genes. **a** RARE-Luciferase activity, **b** qPCR analysis of *Rarβ* expression of NIH3T3 treated with 1 μM RA (*blue*) or 1 μM RA and 100 nM TSA (*green*) (n = 4 each). **c, d** *Rarβ*-promoter ChIP analysis from E8.75 RARE-LacZ mouse embryos with antibodies against Rarα **c** or Rere, Wdr5, Hdac1, and Hdac2 **d**. *Top panels* **c, d**: RARE element of the *Rarβ* promoter ChIP (n = 3). *Bottom panels*: control ChIP from *Rarβ* promoter upstream region (−3 Kb). **e, f** ChIP analysis of the RARE sequence in the RARE-LacZ reporter with antibodies against Rarα **e** or Rere, Wdr5, Hdac1, and Hdac2 **f** using E8.75 RARE-LacZ mouse embryos (n = 3). **g–i** ChIP analysis of *Rarβ* promoter in NIH3T3 treated or not during 1 h with 1 μM RA using antibodies for Rarα **g**, Rere, Wdr5, Hdac1, and Hdac2 **h**, Pol II **i**. *Top panels* **h, i**: ChIP of the RARE element in the *Rarβ*-promoter (n = 3). *Bottom panels*: control ChIP from *Rarβ*-promoter -upstream region (−3 Kb). **j** ChIP of the *Rarβ* promoter with Pol II antibody using NIH3T3 cells treated with 1 μM RA or 1 μM RA and 100 nM TSA for 1 h (n = 3). **k** ChIP with Pol II antibody from NIH3T3 transfected with siRNA for *Hdac1* and treated with 1 μM RA during 1 h (n = 3). **l–n** Co-immunoprecipitation using NIH3T3 transfected with Flag-Hdac1 **l**, Rere-HA and Flag-Hdac1 **m, n**, with anti-Flag **l, m** or anti-HA **n** antibody and immunoblots (IB) for Flag-Hdac1, Rere-HA, and Rarα. **o** *Rarβ*-promoter ChIP analysis from E8.75 RARE-LacZ mouse embryos with an anti-Ehmt2. *Top panel* **o**: RARE element ChIP of the *Rarβ* promoter (n = 3). *Bottom panel*: control ChIP from *Rarβ* promoter upstream region (−3 Kb). **p** ChIP analysis of the RARE sequence in the RARE-LacZ reporter with an anti- Ehmt2 using E8.75 RARE-LacZ mouse embryos (n = 3). Data represent mean ± s.e.m. unless otherwise specified. NS—not significant, *P < 0.05 and **P < 0.01. Student's unpaired two-tailed t-test

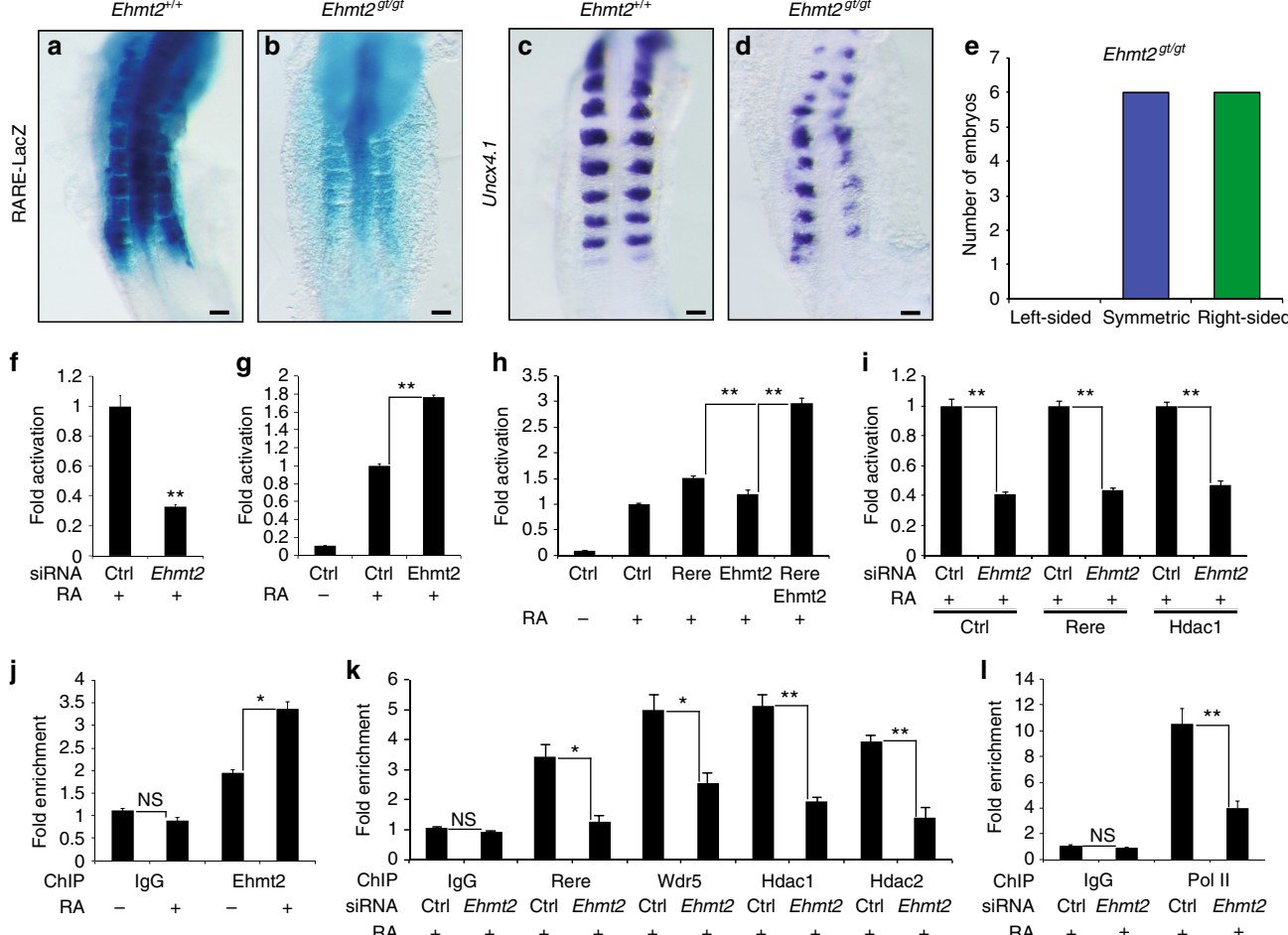

**Fig. 5** Ehmt2 controls retinoic acid signaling and symmetric somite formation. **a, b** RARE-LacZ activity in control *Ehmt2*[+/+] **a** and *Ehmt2*[gt/gt] **b** embryos at E8.75-E9.0 (*dorsal views*). **c, d** In situ hybridization for *Uncx4.1* in wild-type *Ehmt2*[+/+] **c** and *Ehmt2*[gt/gt] **d** embryos at E8.75-E9.0 (*dorsal views*). **e** Graph representing the number of 7- somite to 15-somite stage *Ehmt2*[gt/gt] embryos with left-sided (*orange*), symmetric (*blue*), or right-sided (*green*) delay in somite formation. **f–i** RARE-Luciferase activity from NIH3T3 cells treated with or without 1 μM RA for 20 h. **f** Cells treated with siRNA for *Ehmt2* (*n* = 4). **g** Cells transfected with an Ehmt2 expression plasmid (*n* = 4). **h** Cells co-transfected with Rere and Ehmt2 expression plasmids (*n* = 4). **i** Cells overexpressing Rere or Hdac1 and treated with siRNA against *Ehmt2* (*n* = 4). **j** ChIP of the *Rarβ* promoter with a specific antibody for Ehmt2 using NIH3T3 cells treated or not with 1 μM RA during 1 h (*n* = 3). **k, l** ChIP analysis of the *Rarβ* promoter in NIH3T3 cells transfected with siRNA for *Ehmt2* and treated with 1 μM RA during 1 h. ChIP was performed with antibodies specific to Rere, Wdr5, Hdac1, and Hdac2 **k** or Pol II **l** (*n* = 3). In all graphs data represent mean ± s.e.m. NS—not significant, *$P < 0.05$ and **$P < 0.01$. Student's unpaired two-tailed *t*-test. For each genotype at least 5–10 embryos were analyzed. Bar = 100 microns

generally associated to transcriptional repression[34–36]. However, Rere was shown to act as a coactivator for RA signaling[8] and Hdac1/Hdac2 are required for transcriptional activation of the MMTV promoter downstream of the glucocorticoid receptor[29]. Ehmt2 has also been shown to participate in the positive regulation of a subset of glucocorticoid receptor-regulated genes[37, 38]. Our data show that in the context of RA signaling these transcriptional repressors can form a complex, which acts as a coactivator of the pathway. We further show that the acetyl transferase Ep300 can reduce Rere-dependent RA signaling. This raises the possibility that in this context, Ep300-dependent acetylation of the retinoic acid receptors decreases their transcriptional activity, an effect which could be reversed by Hdac1/2 mediated deacetylation. Thus, studies like those reported here may be necessary to fully understand the functions of co-regulators, which may act as coactivators or corepressors depending on the developmental context and genes involved. The positive role of these proteins in signaling mediated by other nuclear receptors suggests that the WHHERE complex could also have a broader role as a coactivator downstream of nuclear receptor signaling. The positive regulation of RA signaling by

Hdac1 and Hdac2 identified in this work might have significant implications for the potential use of HDAC inhibitors for the treatment of RA-sensitive cancers[39].

## Methods

**Generation of transgenic mice**. The Rere-HA construct was generated by adding two HA tags at the C-terminal extremity of Rere. Rere-HA was cloned downstream of a loxP-STOP-loxP cassette and inserted into the *Rosa26* locus by homologous recombination in 129S6/SvEvTac-derived W4 embryonic stem (ES) cells[40]. Animals were maintained on a 50% C57BL/6 genetic background.

Wdr5 conditional allele (*Wdr5*[fl/+]) mice were generated by the Mouse Clinical Institute (Illkirch-Graffenstaden) by inserting two loxP sites flanking the region encompassing exon 2 and exon 4 in C57BL/6N-derived ES cells (Supplementary Fig. 4a). The Neo selection cassette flanked by FRT site was removed in vivo using a FlpO deleter mice[41]. Animals were maintained on a C57BL/6 genetic background.

**Mice breeding and generation of mutant embryos**. To remove the STOP cassette from the RS-Rere-HA mouse line, we crossed this line with ZP3-Cre mice, which express the Cre recombinase specifically in oocytes[42]. Mice in which the STOP cassette was removed were subsequently intercrossed with *Rere*[om/+] mice[9]. The Rere-HA construct rescued the *Rere*[om/+] mutation indicating that the protein Rere-HA is functional (Supplementary Fig. 1a–c). In order to obtain embryos expressing Rere-HA specifically in the mesoderm for proteomic experiments, we generated a homozygous line for RS-Rere-HA and crossed it with homozygous

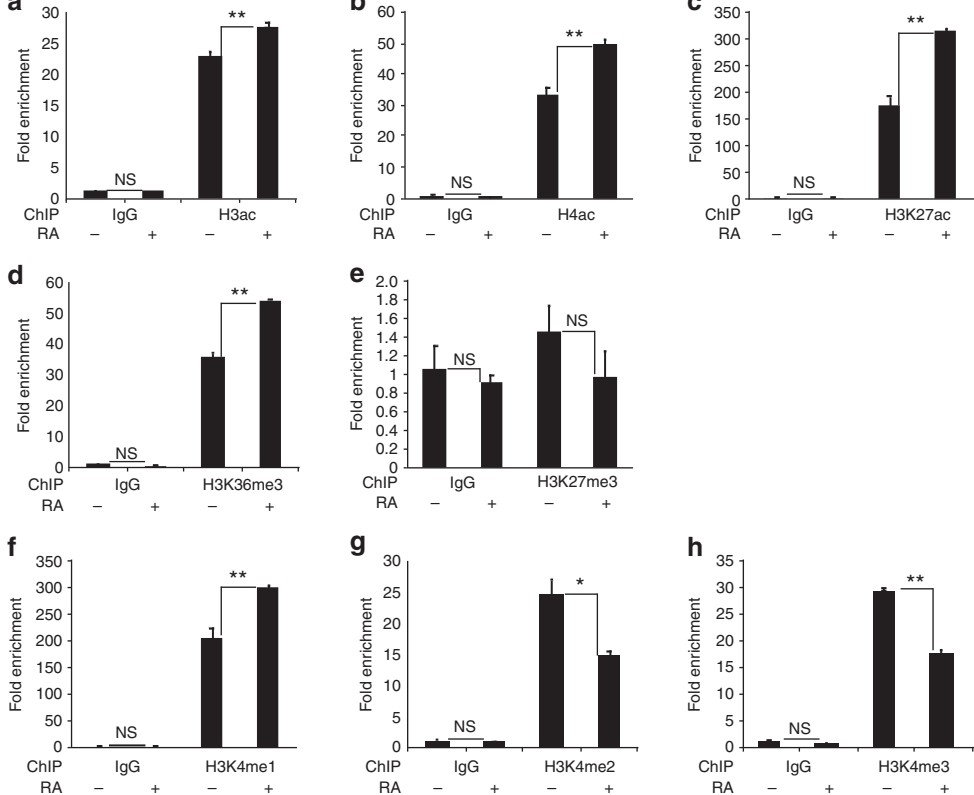

**Fig. 6** Effect of RA treatment on different histone acetylation and methylation marks. **a–h** ChIP analysis of the *Rarβ* promoter from NIH3T3 cells treated with 1 μM RA during 1 h. ChIP was performed with specific antibodies to H3ac **a**, H4ac **b**, H3K27ac **c**, H3K36me3 **d**, H3K27me3 **e**, H3K4me1 **f**, H3K4me2 **g**, and H3K4me3 **h** (*n* = 3). In all graphs data represent mean ± s.e.m. NS—not significant, *$P < 0.05$ and **$P < 0.01$. Student's unpaired two-tailed *t*-test

T-Cre animals[10]. The *Rere*[om/+] mouse line was obtained in an N-ethyl-N-nitrosourea (ENU) screen[9] and the line was maintained on a C57BL/6 genetic background. To analyze conditional mutations of *Wdr5*, we crossed the *Wdr5*[fl/+] mouse line with T-Cre mice. The embryos *Wdr5*[fl/+];*T-Cre* used for analysis only have one mutant allele in the mesodermal tissues. To generate the *Hdac1*[+/−] mouse line, *Hdac1*[fl/fl] animals[21] were crossed with ZP3-Cre mice[42] and maintained on at least 50% C57BL/6 genetic background. Heterozygous males and females *Hdac1*[+/−] were then mated to obtain *Hdac1*[−/−] embryos. The gene-trap allele (gt) *Ehmt2*[gt/+][23] was maintained on at least 50% C57BL/6 genetic background. Embryos *Ehmt2*[gt/gt] were obtained by mating heterozygous males and females for the *Ehmt2* mutation. The mouse mutant for *Ep300*[30] was maintained on at least 50% C57BL/6 genetic background. Heterozygous males and females *Ep300*[+/−] were then mated to obtain *Ep300*[−/−] embryos. To study the status of RA signaling, the RARE-LacZ reporter mice[22] in which LacZ expression is driven by the RARE of the *retinoic acid receptor beta* gene (*Rarβ*) were crossed to: *Rere*[om/+], *Wdr5*[fl/+];*T-Cre*, *Hdac1*[+/−], *Ehmt2*[gt/+], and *Ep300*[−/−] and the mutant embryos were analyzed for β-galactosidase activity using X-gal (5-bromo-4-chloro-3-indolyl-β-D-galactopyranoside). The wild-type and mutant embryos were always stained in the same conditions and same well for 1 h 30 min. All animal procedures were performed according to a protocol approved by IGBMC (#2012-142), in agreement with national and international guidelines.

**Whole-mount in situ hybridization.** Whole-mount in situ hybridizations were performed according to the protocol described in ref. [43]. Briefly, formaldehyde-fixed embryos were treated with protease and refixed with 4% formaldehyde/0.1% glutaraldehyde. Hybridization with DIG-labeled RNA probes was performed under stringent conditions (1.3X SSC, 50% formamide at 65 °C, pH 5) in a buffer containing 0.2% Tween-20 and 0.5% CHAPS. Washed embryos were treated with Boehringer blocking reagent and incubated overnight in alkaline phosphatase-coupled anti-DIG antibody. After extensive washes, embryos were stained from 30 min to 16 h. The probes used are described in the literature: *Uncx4.1*[44], *Lfng*[45], and *Hes7*[46].

**Vector construction and baculovirus generation.** To assay RA signaling in NIH3T3 cells, we used the RA reporter RARE-Luciferase and mouse *Rere-HA* subcloned into pMyc-CMV (Sigma) described in ref. [8]. The mouse full-length *Wdr5*, *Hdac1*, *Hdac2*, and *Rarα* cDNAs were subcloned into pFlag-CMV (Sigma). Human *Flag-Ehmt2* in PCNF plasmid and *Flag-Ehmt1* in pCDNA3 plasmid (both from Nakatani lab) were used to assay RA signaling in NIH3T3 cells. *Rere* N-

terminal region (a.a. 1–571) (N-Rere) and *Rere* C-terminal region (a.a. 572–1558) (Rere C) were subcloned into pMyc-CMV (Sigma) and pFlag-CMV (Sigma), respectively. Human *Hdac1* wild-type (H1-WT) and *Hdac1* mutant (H1-6R) in pEGFP-N1 (Clontech) was a gift from Gordon Hager. Human *Ep300* in pCMVb was a gift from Richard Eckner and mouse *Kat2a* (*Gcn5*) in pCMV-sport2[47] (Invitrogen) (Addgene plasmid 23098) was a gift from Sharon Dent.

For the generation of baculoviruses expressing Flag-Hdac1 and Flag-Hdac2, the full-length cDNA encoding mouse *Hdac1* and *Hdac2* fused with a N-terminal Flag tag, were cloned into the pVL1393 entry vector at BamHI and XbaI sites. For the generation of baculoviruses expressing Rere-Flag, the full-length cDNA encoding mouse *Rere*, was cloned into the pVL1392 entry vector at BplI and KpnI sites. The viruses were then generated from these entry vectors following the transferring genes protocol (Method I) described in Summers and Smith (1987) "A manual of methods for baculo vectors and insect cell culture procedures". The viruses expressing HA-Wdr5 was provided by Dr. Laszlo Tora (IGBMC).

**Cell culture, RA reporter assay and siRNA experiments.** NIH3T3 cells were cultured in DMEM (Invitrogen) containing 4,5 g/l D-glucose, L-glutamine, pyruvate, and 10% newborn calf serum[8]. For RA reporter assay, when cells reached 50% confluence in 24-well plates, they were co-transfected with RARE-pGL4.23 (200 ng), pGL3-Promoter (200 ng), pRL-CMV (5 ng) (Promega), TA-Luc (200 ng) (Panomics) and the different expression constructs containing *Rere*, *Wdr5*, *Hdac1*, *Hdac2*, *Ehmt1*, *Ehmt2*, *N-Rere* (Rere N-terminal region), *Rere C* (Rere C-terminal region), *Hdac1* wild-type (H1-WT), *Hdac1* mutant (H1-6R), *Ep300*, *Rarα* and *Kat2a* using Fugene 6 (Roche). For single transfection, we used 200 ng of each construct whereas in co-transfection experiments 150 ng of each plasmid was used. One micro molar RA (Sigma) was added after 24 h of transfection, and 20 h later, cells were collected and lysed. Renilla luciferase and firefly luciferase activity were measured using a dual luciferase assay kit (Promega) and a microplate luminometer (Berthold Centro XS3 LB 960). Five different histone deacetylase inhibitors were used: Trichostatin A (30–100 nM), Sodium butyrate (1.5–3 mM), Apicidin (300 nM) (all from Sigma) and LAQ824 (60 nM), Panobinostat (30 nM) (both from Selleckchem). UNC0638 and UNC0646 (both from Tocris) were used to inhibit Ehmt2 methyltransferase activity.

For siRNA experiments, NIH3T3 cells were transfected with ON-TARGETplus Smartpool mouse siRNA (Dharmacon) or ON-TARGETplus non-targeting control siRNA pool (Dharmacon) using DharmaFECT 1 or DharmaFECT Duo (for luciferase experiments) reagents (Dharmacon) and following the manufacturer's instructions. The ON-TARGETplus Smartpool mouse siRNA were used against the

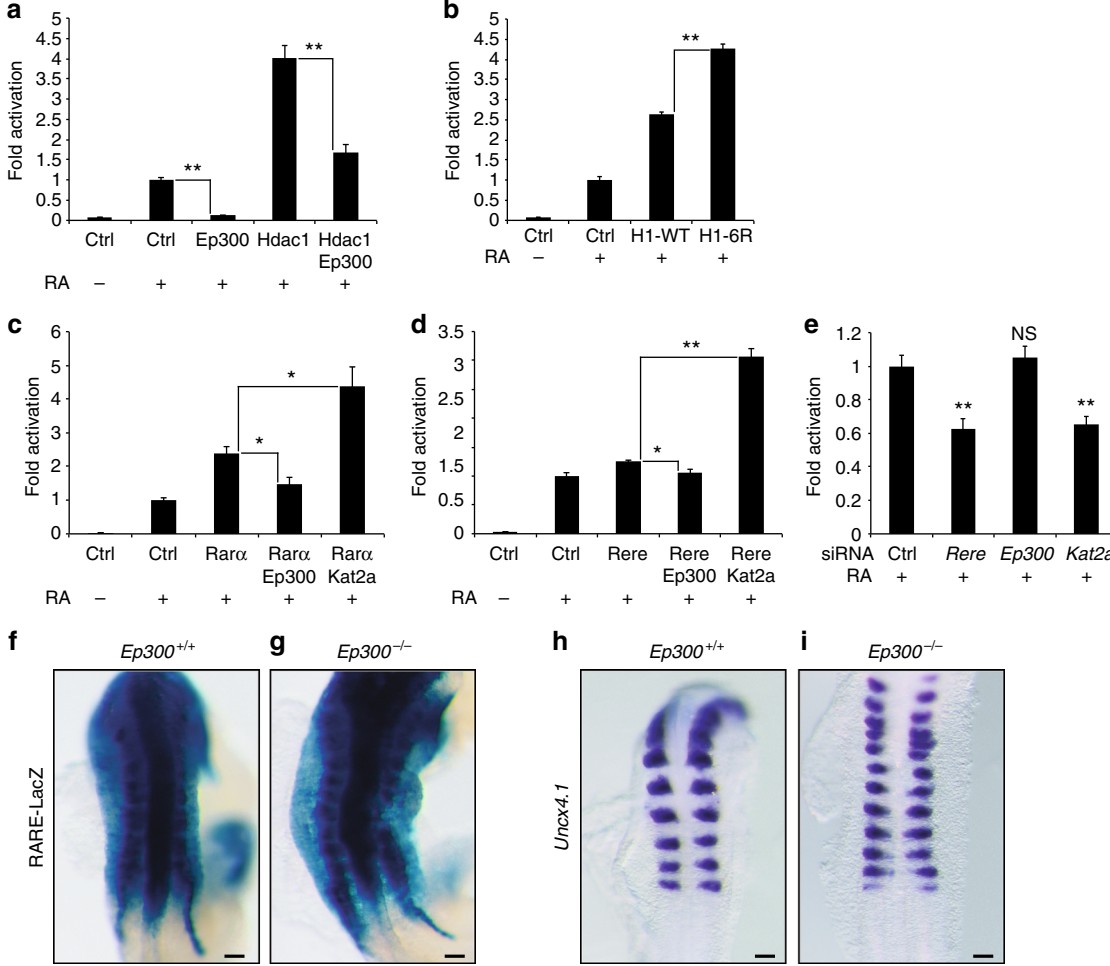

**Fig. 7** Kat2a but not Ep300 acts as a coactivator for retinoic acid signaling. **a–e** RARE-Luciferase activity from NIH3T3 cells treated or not with 1 μM RA for 20 h. **a** Cells transfected with expression plasmids containing *Ep300*, *Hdac1*, or both (*n* = 4). **b** Cells transfected with expression plasmids containing human *Hdac1* (H1-WT) or *Hdac1* mutant (H1-6R) (*n* = 4). **c** Cells transfected with expression plasmids containing *Rarα*, *Rarα*, and *Ep300* or *Rarα* and *Kat2a* (*n* = 3). **d** Cells transfected with expression plasmids containing *Rere*, *Rere* and *Ep300* or *Rere* and *Kat2a* (*n* = 3). **e** Cells treated either with siRNA for *Rere*, *Ep300*, or *Kat2a* (*n* = 4). In all graphs data represent mean ± s.e.m. NS—not significant, *P < 0.05 and **P < 0.01. Student's unpaired two-tailed *t*-test. **f, g** RARE-LacZ expression in wild-type *Ep300*[+/+] **f** and *Ep300*[−/−] **g** embryos at E8.75-E9.0 (*dorsal views*). **h, i** In situ hybridization showing somites labeled with *Uncx4.1* in wild-type *Ep300*[+/+] **h** and *Ep300*[−/−] **i** embryos at E8.75-E9.0 (*dorsal views*). For each genotype at least 5–10 embryos were analyzed. Bar = 100 microns

following target genes: *Rere, Wdr5, Hdac1, Hdac2, Ehmt2, Ep300,* and *Kat2a*. For single siRNA experiments 100 nM of each siRNA was used for the transfection. In the case of double siRNA transfections 50 nM of each siRNA was used for experiments.

**Co-immunoprecipitation experiments and western blots**. NIH3T3 cells were cultured in 10 cm cell culture plates overnight to 70–80% confluence and then transfected with pFlag-*Hdac1* alone or together with pMyc-*Rere*-HA vectors using Fugene 6 (Roche). NIH3T3 cells were cultured in T-25 flasks overnight to 70–80% confluence and then transfected with the expression vectors using Fugene 6 (Roche). Cells were collected 24 h post-transfection and lysed with a hypotonic gentle lysis buffer (10 mM Tris-HCl pH 7.5, 10 mM NaCl, 10 mM EDTA, 0.5% Triton X-100 and Complete protease inhibitors (Roche)). After the lysis, the NaCl concentration was raised to 150 mM. Soluble extracts, together with either mouse IgG (Upstate), M2 anti-FLAG (Sigma) or anti-HA (Sigma), were incubated with Dynabeads M-280 sheep anti-Mouse IgG (Invitrogen) overnight at 4 °C. Beads were washed 10 times with 50 mM Tris-HCl pH 7.5, 150 mM NaCl and 0.05% Triton-X100, and then bound protein was eluted using Laemmli sample buffer (Sigma) and heated at 100 °C for 5 min. Elutions were resolved on 7.5% polyacrylamide Tris-HCl gels (BioRad) and transferred to polyvinylidene difluoride membrane (Millipore). Blots were treated according to a standard western blot protocol.

The following reagents and antibodies were used: mouse IgG (Upstate), M2 anti-Flag (Sigma), anti-HA (Sigma), Dynabeads M-280 sheep anti-mouse IgG (Invitrogen), Flag-coupled HRP (Sigma), HA-coupled HRP (Roche) and Rarα C-20 (Santa Cruz).

To immunoprecipitate the endogenous WHHERE complex from NIH3T3 cells, whole cell protein extract prepared with Tris-HCl pH 7.5, 20% glycerol, 0.2 M KCl, 2X PIC and 1 mM DTT was mixed with Rere antibody (Santa Cruz, sc-98145) coupled with protein G sepharose beads (protein G sepharose 4 fast flow, GE healthcare) or protein G sepharose beads only (negative control IP) for 1 h rotating at 4 °C. Beads were washed in a buffer containing 150 mM KCl, 25 mM Tris-HCl pH 8.0, 10% glycerol, 5 mM MgCl₂, 0.1% NP-40, 2X PIC, 1 mM DTT and eluted after boiling beads in Laemmli buffer (4% SDS, 20% glycerol, 120 mM Tris-HCl pH 6.8, Bromophenol blue 0.02%) followed by western blot analysis against Rere (Santa Cruz, sc-98145, dilution 1/1000), Hdac1 (Abcam, ab7028, dilution 1/1000), Hdac2 (Abcam, ab7029, dilution 1/1000) and Wdr5 (Millipore 07-706, dilution 1/1000). All uncropped western blots can be found in Supplementary Figs. 7–9.

**Real-time PCR assays**. Total mRNA was isolated from NIH3T3 cells using FastLane Cell One-Step Buffer Set (Qiagen). qPCR experiments were performed using Quantifast SYBR Green RT-PCR Kit (Qiagen) in a LightCycler 480 II System (Roche). The following primers from Qiagen (Qiagen Quantitect Primer Assay) were used—with catalog numbers in brackets: *Rere* (QT01075774); *Wdr5* (QT00111685), *Hdac1* (QT02273418), *Hdac2* (QT01161657), *Ehmt2* (QT00115262), *Rarβ* (QT00151956), and *Rplp0* (QT00249375).

**Chromatin immunoprecipitation assay**. NIH3T3 cells were grown in 15 cm dishes and treated with 1 μM of RA (Sigma) for 1 h before collecting. At 70–80% confluency, cells were crosslinked with 1% formaldehyde at room temperature for 10 min. For ChIP experiments with NIH3T3 cells, 50 μg of chromatin were used per immunoprecipitation. F9 or NIH3T3 cells were grown in 15 cm dishes and

**Table 1 Antibodies used for ChIP**

| Antibody | Catalog number (Company) | Concentration/ChIP |
|---|---|---|
| Rere H-113 | sc-98415 (Santa Cruz) | 10 µg |
| Wdr5 | 07-706 (Millipore) | ~ 5 µg (1:200) |
| Hdac1 C-19 | sc-6298 (Santa Cruz) | 10 µg |
| Hdac2 H-54 | sc-7899 (Santa Cruz) | 10 µg |
| Rarα C-20 | sc-551 (Santa Cruz) | 10 µg |
| Rpb1 (total Pol II) | PB-7C2 (Euromedex) | ~ 3 µg (1:300) |
| Ehmt2 | 07-551 (Millipore) | 10 µg |
| H3K9me1 | ab9045 (Abcam) | 2 µg |
| H3K9me2 | ab1220 (Abcam) | 2 µg |
| H3K4me1 | ab8895 (Abcam) | 2 µg |
| H3K4me2 | ab32356 (Abcam) | 2 µg |
| H3K4me3 | ab8580 (Abcam) | 2 µg |
| H3K27ac | ab4729 (Abcam) | 2 µg |
| H3ac | 06-599 (Millipore) | 2 µg |
| H4ac | 06-598 (Millipore) | 2 µg |
| H3K36me3 | ab9050 (Abcam) | 2 µg |
| H3K27me3 | ab6002 (Abcam) | 2 µg |

treated with 0.1 µM and 1 µM of RA (Sigma) 20 h before collecting. At ~80% confluency, cells were crosslinked with 1% formaldehyde at 37 °C for 15 min. Crosslinking was quenched by the addition of glycine to a final concentration of 125 mM for 5 min at room temperature. Cells were lysed in lysis buffer containing 1% SDS, 10 mM EDTA and 50 mM Tris-HCl pH 8.0 and protease inhibitors at a concentration of $3 \times 10^7$ cells/ml. Sonication was performed to produce an average fragment size of ~ 200–1000 bp. Chromatin was diluted fivefold in buffer containing 0.005% SDS, 0.55% Triton X-100, 0.6 mM EDTA, 8.35 mM Tris pH 8.0, 8.35 mM NaCl. Antibodies (5–10 µg per immunoprecipitation) were pre-bound to a protein A sepharose slurry (0.15 g lyophilized protein A sepharose beads (Sigma) were resuspended in 1.3 mL 1X TE with 300 µg sonicated salmon sperm DNA and 1.5 µg BSA). Forty microliters of the resulting slurry was used per immunoprecipitation. Chromatin was pre-cleared using this slurry without antibody and pre-cleared chromatin was then added to the antibody-bead complexes overnight. Chromatin from ~3 × 10⁶ NIH3T3 or F9 cells was used per immunoprecipitation. Complexes were washed the next morning for 10 min each with Buffer A: 0.1% SDS, 1% TritonX-100, 1 mM EDTA, 20 mM Tris-HCl pH 8.0, 50 mM NaCl; Buffer B (same as buffer A except contained 500 mM NaCl); Buffer C: 0.25 M LiCl, 1% NP-40, 1% sodium deoxycholate, 1 mM EDTA, 10 mM Tris-HCl pH 8.0, followed by two washes in 1X TE). Beads were then eluted at room temperature with 1% SDS and 0.1 M NaHCO₃, crosslinks were reversed, proteinase K digestion was performed and DNA was isolated. PCR was performed with the Fast SYBR Green Master Mix (Applied Biosystems) using a 7500 Fast Real-Time PCR System (Applied Biosystems). PCR conditions were as follows: 95 °C (20 s) (1×); 95 °C (3 s) and 60 °C (30 s) (40×). Primer pairs used for the Rar-beta promoter were: 5′-CAGACTGGTTGGGTCATTTGAA-3′ and 5′-GCGAGT-GAACTTTCGGTGAAC-3′. Immunoprecipitated DNA amounts were calculated from a standard curve of input DNA and normalized to rabbit or mouse IgG immunoprecipitations.

For ChIP assay with mouse embryonic tissue, we modified the protocol as follows. E8.75-9.0 embryos were collected from transgenic RARE-LacZ mice in PBS with Complete protease inhibitors (Roche) and crosslinked with 1% formaldehyde at room temperature for 15 min. Crosslinking was quenched by the addition of glycine to a final concentration of 125 mM for 5 min at room temperature. The embryos after dounce homogenization were snap frozen in liquid nitrogen. Homogenized tissue from 24 embryos was suspended in 960 µl of lysis buffer and 40 µl of tissue lysate was subjected to five cycles of sonication in 0.5 ml Protein LoBind Tube (Eppendorf) by Bioruptor (Diagenode) to an average DNA fragment size of 500 bp, and sonicated chromatin was pooled and homogenized, and then divided into 24 specimens to a concentration of one embryo per specimen. Twenty microgram of chromatin were used per immunoprecipitation. A volume of 50 µl of Dynabeads M-280 Sheep anti-Rabbit or Mouse IgG (Invitrogen) was coupled with 5 µg of antibodies in blocking solution (0.1% SDS, 1% TritonX-100, 2 mM EDTA, 20 mM Tris-HCl pH 8.0, 150 mM NaCl, 0.1 mg/ml sonicated salmon sperm DNA and 0.2 mg/ml BSA) at room temperature for 1 h and was used per immunoprecipitation. Antibody-Protein complex was generated in blocking solution at 4 °C overnight and then washed for 5 min each with Buffer D: 0.1%SDS, 1% TritonX-100, 2 mM EDTA, 20 mM Tris-HCl pH 8.0, 150 mM NaCl, 10 mg/ml; Buffer E (same as buffer D except contained 500 mM NaCl); 1X TE containing 0.1% SDS, and all solutions contain 10 mg/ml Linear Acrylamide. Beads were eluted at 55 °C for 1 h with Elution Buffer (1% SDS, 10 mM EDTA, 50 mM Tris-HCl pH 8.0, 1 mg/ml Proteinase K and 10 mg/ml Linear Acrylamide) and then heated at 95 °C for 15 min. Eluted DNA was purified with QIAGEN PCR purification kit. PCR was performed with the Fast SYBR Green Master Mix (Applied Biosystems) using a 7500 Fast Real-Time PCR System (Applied Biosystems). PCR conditions were as follows: 95 °C (20 s) (1×); 95 °C (3 s) and 60 °C (30 s) (40×). Primer pairs used for the RARE-LacZ reporter were: 5′-TGCTGCACGCGGAAGA-3′ and 5′-CCACCAATCCCCATATGGAA-3′.

Antibodies used for ChIP are listed in Table 1.

qPCR was performed with LightCycler 480 SYBR Green I Master mix (Roche) in a LightCycler 480 II System (Roche) using the primers described in Supplementary Table 1.

**Proteomic approach to identify Rere-associated proteins**. To identify proteins binding to Rere, we first generated a mouse colony homozygous for both the T-Cre transgene and the RS-Rere-HA allele. Thus, all the embryos generated by crossing these mice express Rere-HA only in mesodermal tissues. We collected around 600 E10.5 embryos and prepared whole cell extracts using either a low salt (350 mM NaCl) or a high salt (420 mM NaCl) extraction buffer (Supplementary Note 1). For the low salt condition, affinity purification was done at 150 mM NaCl; whereas for the high salt protocol, it was performed at 300 mM NaCl. Each affinity purification was performed on biological replicates of protein extracts from about 50–70 embryos prepared on different days. Two different anti-HA antibodies from Sigma and Roche were used for the affinity purification. For each of the conditions, six elutions were performed with a HA peptide, and the last elution with SDS 2% to completely remove all the proteins attached to the beads. Samples were analyzed by MudPIT as follows: elution 1 run alone, elution 2 and 3 combined and elutions 4 to 7 combined (Supplementary Note 1). A total of 11 and 12 MudPIT analyses were performed for the low and high salt conditions, respectively (Fig. 1a, b and Supplementary Fig. 2d–f). The same purification strategy was applied to embryos from wild-type females to use as a negative control. Proteins were extracted in either low or high salt conditions from 108 × 2 and 100 × 2 embryos prepared on different days, then split in half and affinity purified using either the Sigma or Roche anti-HA antibodies; elutions were combined into 3 digested samples as described above and analyzed independently for a total of 12 MudPIT analyses.

**Multidimensional protein identification technology**. TCA-precipitated protein samples from mouse HA immunoprecipitations were solubilized in either 30 µl or 60 µl (for pooled elutions only) of freshly made 0.1 M Tris-HCl pH 8.5, 8 M urea, 5 mM TCEP (Tris(2-Carboxylethyl)-Phosphine Hydrochloride, Pierce). After 30 min at room temperature, freshly made 0.5 M IAM (Iodoacetamide, Sigma) was added to a final concentration of 10 mM, and the samples were left at room temperature for another 30 min in the dark. Endoproteinase Lys-C (Roche) was first added at an estimated 1:100 (wt/wt) enzyme to protein ratio, for at least 6 h at 37 °C. Urea was then diluted to 2 M with 0.1 M Tris-HCl pH 8.5, CaCl₂ was added to 0.5 mM and modified trypsin (Promega), 1:100 (wt/wt), was added for over 12 h at 37 °C. All enzymatic digestions were quenched by adding formic acid to 5%.

Each trypsin-digested sample was analyzed independently by Multidimensional Protein Identification Technology (MudPIT)[11, 48]. Peptide mixtures were pressure-loaded onto a 100 µm fused-silica column pulled to a 5 µm tip using a P 2000 CO₂ laser puller (Sutter Instruments). The microcapillary columns were packed first with 8 cm of 5 µm C18 reverse phase (RP) particles (Aqua, Phenomenex), followed by 3 cm of 5 µm strong cation exchange material (Partisphere SCX, Whatman), and by 2 cm of RP particles[49]. Loaded microcapillaries were placed in line with LTQ linear ion trap mass spectrometers (Thermo Scientific, San Jose, CA, USA) interfaced with quaternary Agilent 1100 quaternary pumps (Agilent Technologies, Palo Alto, CA, USA). Overflow tubing was used to decrease the flow rate from 0.1 ml/min to about 200–300 nl/min. During the course of a fully automated chromatography, ten 120-minutes cycles of increasing salt concentrations followed by organic gradients slowly released peptides directly into the mass spectrometer[12]. Three different elution buffers were used: 5% acetonitrile, 0.1% formic acid (Buffer A); 80% acetonitrile, 0.1% formic acid (Buffer B); and 0.5 M ammonium acetate, 5% acetonitrile, 0.1% formic acid (Buffer C). The last two chromatography steps consisted in a high salt wash with 100% Buffer C followed by the acetonitrile gradient. The application of a 2.5 kV distal voltage electrosprayed the eluting peptides directly into the mass spectrometers equipped with a nano-LC electrospray ionization source. Each full MS scan (from 400 to 1600 m/z) was followed by five MS/MS events using data-dependent acquisition where the first most intense ion was isolated and fragmented by collision-induced dissociation (at 35% collision energy), followed by the second to fifth most intense ions.

RAW files were extracted into ms2 file format[50] using RAW_Xtract (J.R. Yates, Scripps Research Institute). MS/MS spectra were queried for peptide sequence information on a 157-node dual processor Beowulf Linux cluster dedicated to SEQUEST analyses[51]. MS/MS spectra were searched without specifying differential modifications against a protein database consisting of 73626 non-redundant mouse proteins (NCBI, 2006-09-05 release), 177 sequences for usual contaminants (such as keratins, proteolytic enzymes…), as well as the sequence for mouse Rere with two HA tags at the C-terminus. In addition, to estimate false discovery rates, each non-redundant protein entry was randomized. The resulting "shuffled" sequences were added to the database and searched at the same time as the "forward" sequences. To account for carboxamidomethylation by IAM, +57 Da were added statically to cysteine residues for all the searches.

Results from different runs were compared and merged using CONTRAST[52]. Spectrum/peptide matches were only retained if peptides had to be at least seven amino acids long and be fully tryptic. The DeltCn had to be at least 0.08, with

minimum XCorrs of 1.8 for singly- charged, 2.0 for doubly- charged, and 3.0 for triply-charged spectra, and a maximum Sp rank of 10. Finally, combining all runs, proteins had to be detected by at least 2 such peptides, or 1 peptide with two independent spectra. Proteins that were subset of others were removed.

NSAF7 (Tim Wen, Stowers Institute) was used to create the final report on all detected proteins across the different runs, calculate their respective Normalized Spectral Abundance Factor (NSAF) values and estimate false discovery rates (FDR).

Spectral FDR is calculated as:

$$\mathrm{FDR} = \frac{2 \times \mathrm{SHUFFLED\ Spectral\ Counts}}{\mathrm{Total\ Spectral\ Counts}} \times 100.$$

Protein level FDR is calculated as:

$$\mathrm{Protein\ FDR} = \frac{\mathrm{SHUFFLED\ Proteins}}{\mathrm{Total\ Proteins}} \times 100.$$

Under these criteria the final FDRs at the protein and spectral levels were 1.9% and 0.14% ± 0.085, respectively.

To estimate relative protein levels, NSAFs were calculated for each non-redundant protein, as described in ref. [12, 53, 54]:

$$(\mathrm{NSAF})_i = \frac{(\mathrm{Spectral\ Count/Length})_i}{\sum_{k=1}^{N} (\mathrm{Spectral\ Count/Length})_k}.$$

**WHHERE complex purification and gel filtration analysis**. Protein samples containing Rere-Flag, Flag-Hdac1, Flag-Hdac2, and HA-Wdr5 analyzed by gel filtration, were purified by anti-Flag immunoprecipitation (Flag-IP) from insect cells SF21 (strain *Spodoptera fugiperda*) co-expressing the four genes using a baculovirus expression system, each one cloned into the baculovirus transfer vector pVL1392 or pVL1393. For immunoprecipitation of the WHHERE complex from SF21 cells, cells co-expressing Rere-Flag, Flag-Hdac1, Flag-Hdac2, and HA-Wdr5 were lyzed with a buffer containing 0.4 M KCl, 25% glycerol, 20 mM Tris-HCl pH 8.0, 5 mM MgCl₂, 0.5% NP-40, 2X complete protease inhibitor cocktail EDTA-free (Roche) and 1 mM DTT. After centrifugation of the lysate for 10 min at 8000 r.p.m. at 4 °C, the supernatant containing the whole cell protein extract was collected and incubated with anti-Flag antibody coupled with agarose beads (anti-Flag M2 affinity gel, Sigma) for 1 h at 4 °C, to perform the Flag-IP. After washing beads in a buffer containing 150 mM KCl or 500 mM KCl, 25 mM Tris-HCl pH 8.0, 10% glycerol, 5 mM MgCl₂, 0.1% NP-40, proteins were eluted by peptide binding competition using the Flag-tag peptide DYKDDDDK (2 mg/ml final) dissolved into a buffer containing 100 mM KCl, 25 mM Tris-HCl pH 8.0, 10% glycerol, 5 mM MgCl₂, 0.1% NP-40. Proteins eluate was then concentrated on amicon-ultra centrifugal filter units (Millipore, membrane cutoff of 30 kDa) before gel filtration analysis. To study the direct protein–protein interaction between Rere and Wdr5, Rere-Flag, and HA-Wdr5 were co-expressed and Flag-tagged purified as described above. The gel filtration analysis of the purified complexes was performed on a Superose 6 PC 3.2/30 (GE Healthcare) mounted on an AKTA micro chromatography system (GE Healthcare) using a buffer containing 25 mM Tris-HCl pH 8.0, 100 mM KCl, 5 mM MgCl₂, and 2% glycerol. Calibration of the column was performed using a standard calibration kit (BioRad), containing Thyroglobulin (670 kDa), ɣ-globulin (158 kDa), ovalbumin (44 kDa), myoglobulin (17 kDa), and Vitamin B12 (1355 Da). The exclusion volume $V_0$ was determined using calf thymus DNA (> 2000 bp), while the total volume of the column was measured using L-Tyrosine. For the experiment, 80–100 μl of purified sample was injected through an injection loop to the column. Flow rate was set to 40 μl/min and SEC fractions of 75 μl each were collected throughout the gel filtration run and analyzed by western blotting to check for the presence of Rere, Hdac1, Hdac2, and Wdr5. The following antibodies were used for western blot: Rere (Santa Cruz, sc-98415), Hdac1 (Abcam, ab7028), Hdac2 (Abcam, ab7029), and Wdr5 (Abcam, ab22512).

**Statistics**. Statistics were performed using Excel software (Microsoft). Significance of difference was analyzed with a Student's unpaired two-tailed *t*-test. All data are presented as mean ± s.e.m. and a *P*-value of 0.05 was considered significant. The results were presented as the average of at least three independent experiments unless otherwise stated in the legends.

**Data availability**. The authors declare that all data supporting the findings of this study are available within the article and its supplementary information files or from the corresponding author upon reasonable request. The mass spectrometry data set has been deposited in the MassIVE/ProteomeXchange under accession codes MSV000080920 and PXD006303.

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

## Acknowledgements

We thank members of the Pourquié laboratory for discussions and comments on the manuscript. We are grateful to R. Schneider, J. Workman, M.E. Torres-Padilla and L. Tora for critical reading of the manuscript. We thank A. Peterson, J. Rossant, E. Olson, R. Feil, M. Lewandoski and Andrew Kung for providing mouse lines. We thank Gordon Hager, Sharon Dent and Richard Eckner for sharing reagents. We thank the Stowers Institute and IGBMC core facilities, particularly the Baculovirus (I. Kolb-Cheynel), Structural Biology (I. Billas) and Cell Culture facilities. We are particularly thankful to M.C. Birling and the Institute Clinique de la Souris for the generation of conditional mouse lines and for animal care (R. Bour and S. Untereiner). The research was supported by the Howard Hughes Medical Institute, the Stowers Institute for Medical Research and the European Research Council (ERC advanced grant to O.P.).

## Author contributions

G.C.V.-N. designed, performed and analyzed the experiments with O.P. G.C.V.-N., M.F. and J.M.G. performed the molecular biology, biochemistry, and gel filtration experiments. G.C.V.-N., J.-L.P. and M.M. did the qPCR and qChIP experiments. G.C.V.-N., M.E.S., A.S., L.F. and M.P.W. performed and analyzed the proteomic experiment. G.C.V.-N. did the mouse analysis. G.C.V.-N. and O.P. wrote the manuscript and supervised the project. All authors discussed and agreed on the results and commented on the manuscript.

## Additional information

**Competing interests:** The authors declare no competing financial interests.

