## [Peer Review File · Nature Communications]

Reviewers' Comments:

Reviewer #1 (Remarks to the Author)

This manuscript describes a very important advance in our understanding of how transcriptional coregulators interact to regulate retinoic acid (RA) signaling, focusing in particular on how RA controls formation of somites from presomitic mesoderm. Although RA is known to be required for bilateral symmetry during somitogenesis, the mechanism has remained very unclear other than the observation that the coregulator Rere is also required as previously reported by the authors. Here, the authors performed an impressive proteomic analysis to identify Rere-binding proteins in embryonic mesoderm using mouse embryos expressing HA-tagged Rere in posterior mesoderm which gives rise to somites. They found that Rere co-immunoprecipitated with four other important coregulators (Wdr5, Hdac1, Hdac2, and Ehmt2) - this result provides valuable insight as these are complexes that form *in vivo*. The authors go on to show that mouse embryos deficient for either Rere, Wdr5, Hdac1, or Ehmt2 exhibit significant reductions in RA signaling and somite asymmetry. They also showed with ChIP that each of these coregulators can localize near the RA response element for the RAR β gene which encodes one of the RA receptors; siRNA knockdown showed that they all can contribute to RA activation of the RAR β promoter. A very interesting point that the authors mention is that although Wdr5 is known to function as a coactivator, many studies show that the other proteins function as corepressors. As critical evidence was provided *in vivo*, I agree that the authors can make the argument that all of these proteins act together as coactivators, at least for the RAR β promoter. Overall, these studies break new ground by showing that *in vivo* studies can challenge the current view (mostly from *in vitro* and cell line studies) of how we determine whether a transcriptional coregulator functions as a coactivator or a corepressor.

Specific Points:

1. Page 2 (line 44) - it would be good to also add a reference for a more recent review on RA signaling (Cunningham, Nature Rev Mol Cell Biol, 2015).
2. Page 2 (line 45) - add references for RA corepressors and coactivators.
3. Page 3 (line 65) - to enable the reader to quickly understand the scope of the *in vivo* proteomic study, add "from approximately 600 T-Cre;RS-Rere-HA embryos".
4. Page 6 (line 135) - it should be mentioned that no change in Lfng and Hes7 suggests that loss of Wdr5 does not affect the somite clock.
5. Page 9 (Line 216) - as the addition of Hdac inhibitors reduced RAR β gene activation it is reasonable to suggest that Hdac1/2 may act on non-histone substrates to activate RAR β , but it could be mentioned that enzyme activity may not be needed as it cannot be ruled out that these inhibitors might disrupt binding of Hdac1/2 in the complex.
6. Page 10 (line 238) - it would be useful to mention that RA has been shown to directly repress caudal Fgf8 transcription and that RA stimulates recruitment of Hdac1 and PRC2, but not Rere, to a RARE upstream of Fgf8 in mouse embryos (ref. 23). In the case of the Fgf8 RARE, Hdac1 appears to be acting as a corepressor recruited when RA is present, while Rere may act as a coactivator that is removed from the RARE when RA is present. Thus, it would be interesting in the future to determine if WHHERE or some of its components function as RA-dependent corepressors for the Fgf8 promoter.
7. Page 10 (line 240) - as Rere, Hdac1/2 and Ehmt2 are viewed as corepressors by many in the epigenetics field, it may be useful to add that *in vivo* studies like those reported here may be

necessary to fully understand the functions of coregulators which may act as coactivators for some genes and corepressors for others.

8. Methods (line 35) - in order to show the reduction in RA activity for the various mutants carrying RARE-lacZ, how long were the wild-type and mutant embryos stained?

Reviewer #2 (Remarks to the Author)

Summary of the key results:

A mammalian complex consisting of Wdr5, Hdac1, Hdac2, and Rere is identified, and reported to recruit Ehmt2 to enhance activity of the RAR β promoter. Mouse models are used to link this protein complex to the embryonal bilateral symmetry.

Originality and interest:

The identification of a mammalian Wdr5, Hdac1, Hdac2, and Rere complex extends earlier findings in *Drosophila* - a non-mammalian species (Wang L, Charroux B, Kerridge S, Tsai CC. *EMBO Rep.* 2008 Jun; 9(6):555-62), and adds Wdr5 to the complex. The findings are relevant for understanding the relation between transcriptional activity and animal development, but do not provide a major leap forward. Also, the positive regulation by Hdac1/2 has relevance for cancer therapy, yet in light of the overwhelming amount of published data against, this regulation is not convincingly demonstrated.

Data & methodology:

Multiple very elegant mouse models (Rere-om, Wdr-fl, Hdac-ko) are used to address the biological relevance of the identified components in the context of developing somites. The use of mesoderm specific T-cre regulated expression provides a highly appropriate approach for tissue specific knockout.

The immune-precipitation of the complex using a flag-tag (Figure 1c) is highly inappropriate since multiple proteins in the proposed complex are flag-tagged. This approach obscures any interpretation of the data.

Reporter assays are generally useful for evaluation of multiple parameters, but assays using a control construct with a mutated RARE are missing.

Conclusions:

Hdac1/2 and Ehmt2 are reported to enhance activity of the RAR β promoter. However, because of their histone deacetylating capabilities, Hdacs generally function as negative regulators of transcription. Specifically, Hdac1 is repressing RAR β function in embryonic stem cells, and transcriptional activation is associated with removal of Hdac1 (Urvalek and Gudas. *J Biol Chem.* 2014 Jul 11; 289(28):19519-30). In addition, the recruitment by Rere of Hdac1, Hdac2, and Ehmt2 was demonstrated to repress gene activity in *Drosophila* (Wang L, Charroux B, Kerridge S, Tsai CC. *EMBO Rep.* 2008 Jun; 9(6):555-62). The recruitment of histone acetyl transferases and acetylation of histones associated with activation of retinoic acid responsive genes further corroborate the repressive functions of Hdacs. It is noted, that Hdac1/2 may act on unspecified non-histone substrates, but why Hdac1/2 would not target the histone substrate is not discussed.

Overall, in spite of the elegant Hdac mouse model, and the extensive use of Hdac inhibitors, the positive regulation by Hdac1/2 is not convincingly demonstrated.

Suggested improvements:

Address if endogenous Hdac1/2 (and Wdr5) can be immune-precipitated from NIH3T3 cells using a Flag-tagged Rere. This would address the possibility that the Rere/Hdac1/Hdac2/Wdr5 association could represent multiple complexes, e.g. Rere/Wdr5, Hdac1/Wdr5, and Hdac2/Wdr5 (as suggested by the 1:1:1:3 stoichiometry and by the delayed elution of Hdac1 and Hdac2 relative to Rere, Figure 1f).

The Chip-assays of histone mark association with RAR β should be included in the main manuscript.

The reporter assay in NIH3T3 cells should include a construct with a mutated RARE. This would address if the signaling is a generic reporter artefact, or indeed through the RARE.

The Hdac1/2 association with the RAR β promoter was assessed after only 1 hr of RA treatment (Figure 4), which is before transcript levels increase. If high Hdac1/2 levels are maintained at later time points this could indeed support the controversial function of Hdac1/2 as a coactivator.

References: Overall appropriate references are cited, but are on occasions presented in a way which overstates agreement with and/or oversells the novelty of the current findings. As an example, a complex of Rere/Hdac1/Hdac2/Ehmt2 was already identified in *Drosophila* (as is correctly noted), but was reported to function by repressing transcription (Wang L, Charroux B, Kerridge S, Tsai CC. *EMBO Rep.* 2008 Jun; 9(6):555-62). It is puzzling why this complex would be repressing in *Drosophila*, while activating in mouse - particularly since most of the components as stated in the manuscript have known repressive functions.

Reviewer #3 (Remarks to the Author)

Vilhais-Neto et al., have undertaken a proteomic approach to discover a coactivator complex that regulates retinoic acid (RA) signalling. Core epigenetic regulators (histone deacetylases and a histone methyl transferase) has been discovered to be part of this coactivator complex. They have followed this with a biochemical and cell-based analysis of its effects on RA-mediated transcription. This characterization appears to be detailed and thorough.

Mutations that target some of the new constituents were observed to disrupt left-right symmetries during somitogenesis in mouse embryos. However, there is no information about whether this complex is active in other embryonic tissues or whether it is a feature only of the paraxial mesoderm. Thus, even though the authors propose a link between epigenetic regulators and embryonic symmetry, it is unclear to what extent the effects of the mutations are specific towards embryonic symmetry. Overall, the paper does not yield any new insights into the mechanisms by which left-right symmetry is established.

The following comments need to be addressed:

1) Histone deacetylases and methyl transferases are ubiquitous long-lived epigenetic proteins involved with many different transcription factors. And, as the authors have mentioned, Rere is already known to interact with hdac1 and hdac2. Is it therefore an expected result that a third epigenetic regulator (Ehmt2) is also a part of the complex? How does this significantly increase our understanding of RA control of transcription?

2) Both Fig. 2c and 2d display activation upon Hdac2 siRNA, but they seem to have different levels of fold activation. Can the authors explain or clarify this?

3) The authors observe a dramatic phenotype upon crossing Rere mutants with Wdr5-TCre, where 35% of the embryos lacked segmented structures on the right side. How do these mutations lead to much severe phenotypes than RA synthesizing mutants? Does this suggest that Wdr5 and Rere are likely to be involved in other signalling pathways regulating embryonic symmetry?

4) It is unclear how any arguments on symmetry versus asymmetry can be made from the phenotype of Wdr5 mutants (Fig. 3h), which have an overall altered embryonic morphology and highly disrupted somitogenesis. From the data presented, one can conclude that somitogenesis is disrupted, but one cannot conclude that this is a specific phenotype, i.e. that somites are affected independently of any other structure, nor can one conclude that symmetry is affected.

5) Can the authors comment on why Rara was not found to be increased on the Rar β promoter in the ChIP experiments (Fig. 4g,h) performed after 1 hr of RA treatment?

6) Based on the observed pull-down of Hdac1 and Rara (Fig. 4l-n), the authors suggest that Hdac1 might connect the WHHERE complex to Rara, which seems a pretty strong statement. The data show that the proteins can be found in in the same complex, and nothing more.

Reviewer #4 (Remarks to the Author)

The authors of the manuscript entitled: "A new coactivator complex required for retinoic acid-dependent regulation of embryonic symmetry" describe the isolation of a set of proteins which are required for somitogenesis symmetry in mice. They also bring evidence that these proteins bind to the RARE element and induce transcription of a RARE driven reporter gene in the cell and in vivo. Amongst those proteins is WDR5, which is a novel factor involved in RA-driven vertebrate bilateral symmetry. They also present evidence that inhibition of HDACs interferes with bilateral symmetry and that the protein methyltransferase EHMT2 has a positive role in symmetric somite formation. Albeit the in vivo and the functional data presented by the authors are compelling, several aspects of this work require major revision prior publication in this journal.

1) WHHERE complex:

- a) Although the identification of the Rere interaction partners in the transgenic mouse is indisputable, the novelty of the WDR5 binding to Rere (line 75) has to be mitigated since the interaction between WDR5 and Rere has been already published in Hein et al., Cell 2015 (reference to be included).
- b) Also the claim for the direct interaction between WDR5 and Rere (Fig 1d) requires the proper control which is an IP with anti -Flag antibodies from extract containing only HA-WDR5. Moreover, the authors should exclude the presence of endogenous HDAC1/2 in those IPs (Fig 1d).
- c) For the protein association data after co-expression in the baculovirus system (Fig. 1c, 1e and 1f), the author should explain and show the specificity of the M2 -resin pull-down for either HDAC1, HDAC2 or Rere which are all tagged with Flag before concluding on complex formation

a) Stoichiometry of the WHHERE complex components: the authors use spectral counts (NSAF) to calculate stoichiometry. Since spectra counting suffers from a poor precision, there are more advanced methods nowadays to estimate complex composition which I would encourage to apply as the one in published in Hein et al , Cell 163, Issue 3, 2015.

For a precise determination of a the stoichiometry methods e.g. AQUA peptides, SILAC or iTRAQ/TMT are available.

2) Functional assay:

- a) The cellular reporter assay shown in Fig 2 albeit reproducible within the same experiment (very small error bars) suffers from a certain variability between different experiments (i.e. compare Figs. 2a and 2b after HDAC1 overexpression, Figs 2d and 2f for the effect of HDAC1 siRNA or HDAC1 and HDAC2 si RNA after WDR5 overexpression). It is therefore hard to conclude on the effect of the double knock-down of the HDACs if the experiments are not done side by side with the single knock-downs. Alternatively, the statement in line 111 should be removed.
- b) To validate the unexpected transcriptional activation effect of the knock down of HDAC2 (Fig 2c) the authors should show the effect of the HDAC2^{-/-} mice mutant on the expression of the RARE LacZ reporter.
- c) In order to clearly demonstrate an RA-dependent activation role of the WHHERE complex the authors should show reporter data in the absence of RA treatment following overexpression or knock-down of the different proteins (and not just of control). Moreover, they should show the effect of at least 1 example of knock down or overexpression of a known RA activator (i.e. EP300). A combination of the latter with any of the WHHERE complex components, might also give insight into the molecular mechanism of HDAC-mediated increase of expression of RA-dependent genes (indeed HDACs have been already shown to increase gene expression of nuclear receptor controlled genes by acting, amongst others, on PolII recruitment _Welsbie et al 2009)

3) Interaction with EHMT2:

a) The weak retrieval of the EHMT2 protein in the complex might also be due to substoichiometric interaction (to consider in line 187).

In any case, the statement in line 206 on a scaffolding role for EHMT2 for RA regulated genes, should be mitigated since the retrieval of EHMT2 with the WHHERE complex is weak (Fig 5a), and EHMT2 binds to the RarB promoter also in absence of RA (fig 5c). This is also in agreement with no change in the level of H3K9me1 or of H3K9me2 at this promoter. The use of potent and selective G9a inhibitor might also clarify whether the methyltransferase activity of EHMT2 is required for this function (Kubicek S et al. (2007) Reversal of H3K9me2 by a small-molecule inhibitor for the G9a histone methyltransferase. Mol Cell 25: 473-481)

b) The authors should indicate the position of EHMT2 in Suppl fig 1f and/or 1e

Response to the reviewers

Reviewer #1 (Remarks to the Author):

This manuscript describes a very important advance in our understanding of how transcriptional coregulators interact to regulate retinoic acid (RA) signaling, focusing in particular on how RA controls formation of somites from presomitic mesoderm. Although RA is known to be required for bilateral symmetry during somitogenesis, the mechanism has remained very unclear other than the observation that the coregulator Rere is also required as previously reported by the authors. Here, the authors performed an impressive proteomic analysis to identify Rere-binding proteins in embryonic mesoderm using mouse embryos expressing HA-tagged Rere in posterior mesoderm which gives rise to somites. They found that Rere co-immunoprecipitated with four other important coregulators (Wdr5, Hdac1, Hdac2, and Ehmt2) - this result provides valuable insight as these are complexes that form in vivo. The authors go on to show that mouse embryos deficient for either Rere, Wdr5, Hdac1, or Ehmt2 exhibit significant reductions in RA signaling and somite asymmetry. They also showed with ChIP that each of these coregulators can localize near the RA response element for the RAR β gene which encodes one of the RA receptors; siRNA knockdown showed that they all can contribute to RA activation of the RAR β promoter. A very interesting point that the authors mention is that although Wdr5 is known to function as a coactivator, many studies show that the other proteins function as corepressors. As critical evidence was provided in vivo, I agree that the authors can make the argument that all of these proteins act together as coactivators, at least for the RAR β promoter. Overall, these studies break new ground by showing that in vivo studies can challenge the current view (mostly from in vitro and cell line studies) of how we determine whether a transcriptional coregulator functions as a coactivator or a corepressor.

Specific Points:

1. Page 2 (line 44) - it would be good to also add a reference for a more recent review on RA signaling (Cunningham, Nature Rev Mol Cell Biol, 2015).

- The reference has been added to the text.

2. Page 2 (line 45) - add references for RA corepressors and coactivators.

- References have been added to the text.

3. Page 3 (line 65) - to enable the reader to quickly understand the scope of the in vivo proteomic study, add "from approximately 600 T-Cre;RS-Rere-HA embryos".

- The text has been modified to include reviewer suggestion.

4. Page 6 (line 135) - it should be mentioned that no change in Lfng and Hes7 suggests that loss of Wdr5 does not affect the somite clock.

- The text has been modified to include reviewer suggestion.

5. Page 9 (Line 216) - as the addition of Hdac inhibitors reduced RARb gene activation it is reasonable to suggest that Hdac1/2 may act on non-histone substrates to activate RARb, but it could be mentioned that enzyme activity may not be needed as it cannot be ruled out that these inhibitors might disrupt binding of Hdac1/2 in the complex.

- This discussion has now been added in the text.

6. Page 10 (line 238) - it would be useful to mention that RA has been shown to directly repress caudal Fgf8 transcription and that RA stimulates recruitment of Hdac1 and PRC2, but not Rere, to a RARE upstream of Fgf8 in mouse embryos (ref. 23). In the case of the Fgf8 RARE, Hdac1 appears to be acting as a corepressor recruited when RA is present, while Rere may act as a coactivator that is removed from the RARE when RA is present. Thus, it would be interesting in the future to determine if WHHERE or some of its components function as RA-dependent corepressors for the Fgf8 promoter.

- The text has been modified to include the reviewer suggestion.

7. Page 10 (line 240) - as Rere, Hdac1/2 and Ehmt2 are viewed as corepressors by many in the epigenetics field, it may be useful to add that in vivo studies like those reported here may be necessary to fully understand the functions of coregulators which may act as coactivators for some genes and corepressors for others.

- The text has been modified to include reviewer suggestion.

8. Methods (line 35) - in order to show the reduction in RA activity for the various mutants carrying RARE-lacZ, how long were the wild-type and mutant embryos stained?

- The wild-type and mutant embryos were always stained in the same conditions and same well for 1h30. This has now been added to the methods.

Reviewer #2 (Remarks to the Author):

Summary of the key results:

A mammalian complex consisting of Wdr5, Hdac1, Hdac2, and Rere is identified, and reported to recruit Ehmt2 to enhance activity of the RAR β promoter. Mouse models are used to link this protein complex to the embryonal bilateral symmetry.

Originality and interest:

The identification of a mammalian Wdr5, Hdac1, Hdac2, and Rere complex extends earlier findings in *Drosophila* - a non-mammalian species (Wang L, Charroux B, Kerridge S, Tsai CC. EMBO Rep. 2008 Jun;9(6):555-62), and adds Wdr5 to the complex. The findings are relevant for understanding the relation between transcriptional activity and animal development, but do not provide a major leap forward. Also, the positive regulation by Hdac1/2 has relevance for cancer therapy, yet in light of the overwhelming amount of published data against, this regulation is not convincingly demonstrated.

- We understand that our results are surprising and we are aware that in most contexts in which they have been studied, Hdac1/Hdac2 were shown to function as corepressors. However, there are studies describing a coactivator role for Hdac1/Hdac2 for other nuclear

receptors such as the glucocorticoid receptors (Qiu et al. Mol Cell. 2006). Also in yeast, the Hdac1/Hdac2 homologues RPD1 and RPD3 can also function as coactivators (Vidal et al, Mol Cell Biol, 1991).

Much of the work examining Hdac1/Hdac2 function in transcription has however been done in cell lines *in vitro* and its relevance *in vivo* remains to be established. In our study, we demonstrate that Hdac1/Hdac2 function as coactivators of RA signalling using gain and loss function approaches in cell lines as well as *in vivo* analysis in the mouse mutants. All these data clearly establish a coactivator role for these proteins. Nevertheless, in the revised version of the paper we included further experiments strengthening the role of HDACs in the positive regulation of RA signalling:

- Hdac1/Hdac2 have been shown to bind Rere N-terminal region (N-Rere) (Zoltewicz et al. Development 2004; Wang et al. Genes Dev. 2006; Wang et al. EMBO Rep. 2008). We now report that overexpression of N-Rere but not Rere-C strongly activates RA pathway (Fig. 2j). Moreover, we show that activation by N-Rere is HDAC-dependent (Fig. 2k).

- Ep300 has been shown to acetylate Hdac1 to inactivate its deacetylase activity (Qiu et al. Mol Cell. 2006). In agreement with these findings, we now show that Ep300 overexpression inhibits RA signalling and reduced Hdac1 activation of RA signalling (Fig. 7a). Furthermore, using a mutant form of Hdac1 (H1-6R) resistant to Ep300 inhibition, we observed stronger RA pathway activation than with Hdac1 wild-type form (H1-WT) (Fig. 7b). While we agree that this is again an unexpected finding as Ep300 is usually considered a coactivator, our *in vivo* and *in vitro* data strongly argues for its negative role on Hdac1-dependent RA signalling.

Data & methodology:

Multiple very elegant mouse models (Rere-om, Wdr-fl, Hdac-ko) are used to address the biological relevance of the identified components in the context of developing somites. The use of mesoderm specific T-cre regulated expression provides a highly appropriate approach for tissue specific knockout.

The immune-precipitation of the complex using a flag-tag (Figure 1c) is highly inappropriate since multiple proteins in the proposed complex are flag-tagged. This approach obscures any interpretation of the data.

- As mentioned above, the interaction between Rere, Hdac1 and Hdac2 was previously reported in: Zoltewicz et al. Development 2004; Wang et al. Genes Dev. 2006; Wang et al. EMBO Rep. 2008 and our study extends these earlier findings by adding Wdr5 to the list of Rere binding partners and by demonstrating that these four proteins form a complex. The goal of the experiment described in Fig. 1c was to validate the identification of Wdr5, as a novel binding partner of Rere/Hdac1/Hdac2. By performing Flag-immunoprecipitation of Flag-tagged Rere, Hdac1 and Hdac2, we ensured to pull down the three interacting proteins while validating the Wdr5 interaction using a different tag, HA-tag. The existence of the WHHERE protein complex is demonstrated in an independent experiment, using gel filtration and purified proteins as described in Fig. 1e,f. We have also clarified the conclusions in the text.

- In the revised text, we now include results showing no direct interaction between Wdr5 and Hdac1/Hdac2 (Supplementary Fig. 2c) from co-immunoprecipitation experiments performed from baculovirus-insect cells co-expressing HA-Wdr5 and Flag-Hdac1/Flag-Hdac2. This would suggest that Rere binds independently to Wdr5 and to Hdac1/Hdac2.

Reporter assays are generally useful for evaluation of multiple parameters, but assays using a control construct with a mutated RARE are missing.

- The RARE-Luciferase reporter used in this study for cell cultures experiments is the same RA reporter in the mouse line RARE-LacZ (Rossant et al. Genes Dev. 1991) (also used in this study) that has been widely used in RA research projects. This reporter is highly sensitive for RA signalling, different groups have extensively characterized it and it is a standard RA reporter used in cell and developmental biology papers such as Blanco et al. Genes Dev. 1998; Lin et al. Nature 1998; Hoffman et al. J Cell Biol. 2006; Vilhais-Neto et al. Nature 2010; Cunningham et al. Genesis 2011; Papi et al. Cell Death Differ. 2012. Due to the high specificity of the reporter for RA signalling, comparison to the mutated form is not necessary and has not been performed in such studies.

Conclusions:

Hdac1/2 and Ehmt2 are reported to enhance activity of the RAR β promoter. However, because of their histone deacetylating capabilities, Hdacs generally function as negative regulators of transcription. Specifically, Hdac1 is repressing RAR β function in embryonic stem cells, and transcriptional activation is associated with removal of Hdac1 (Urvalek and Gudas. J Biol Chem. 2014 Jul 11;289(28):19519-30). In addition, the recruitment by Rere of Hdac1, Hdac2, and Ehmt2 was demonstrated to repress gene activity in Drosophila (Wang L, Charroux B, Kerridge S, Tsai CC. EMBO Rep. 2008 Jun;9(6):555-62). The recruitment of histone acetyl transferases and acetylation of histones associated with activation of retinoic acid responsive genes further corroborate the repressive functions of Hdacs. It is noted, that Hdac1/2 may act on unspecified non-histone substrates, but why Hdac1/2 would not target the histone substrate is not discussed.

Overall, in spite of the elegant Hdac mouse model, and the extensive use of Hdac inhibitors, the positive regulation by Hdac1/2 is not convincingly demonstrated.

- We respectfully disagree with this referee and we think our data strongly argues for a positive role of Hdac1/Hdac2 in RA signalling. We demonstrate that:

- Hdac1/Hdac2 overexpression *in vitro* leads to an upregulation of RA signalling.
- HDAC inhibition with known inhibitors such as TSA or SB or siRNAs (at least for *Hdac1*) leads to an inhibition of RA signalling.
- Using a similar set of inhibitors or siRNAs, we show that Hdac1 is required for Pol II recruitment to the promoter of the RA target gene *Rar β* .
- *In vivo*, *Hdac1* mutation leads to a decrease in the retinoic acid reporter RARE-LacZ signal in the mouse embryo.
- *Hdac1* mutants phenocopy the somite desynchronization observed in *Raldh2* mutants (which are the strongest mutants in the RA pathway), a phenotype which so far has only been observed in *Raldh2* and *Rere* mutants.

Suggested improvements:

Address if endogenous Hdac1/2 (and Wdr5) can be immunoprecipitated from NIH3T3 cells using a Flag-tagged Rere. This would address the possibility that the Rere/Hdac1/Hdac2/Wdr5 association could represent multiple complexes, e.g. Rere/Wdr5, Hdac1/Wdr5, and Hdac2/Wdr5 (as suggested by the 1:1:1:3 stoichiometry and by the delayed elution of Hdac1 and Hdac2 relative to Rere, Figure 1f).

- By co-immunoprecipitation of endogenous Rere from NIH3T3 cells, we demonstrate that Rere forms a complex with Wdr5, Hdac1 and Hdac2 (Fig. 1g).

- Regarding the existence of multiple complexes (e.g. Rere/Wdr5, Hdac1/Wdr5, and Hdac2/Wdr5), we show that Wdr5 binds directly to Rere (Fig. 1d). Also, we did not observe direct binding of Wdr5 to Hdac1/Hdac2, after co-immunoprecipitation experiments from baculovirus-insect cells co-expressing HA-Wdr5 and Flag-Hdac1/Flag-Hdac2 (Supplementary Fig. 2c). This would suggest that the 1:1:1:3 stoichiometry might be conferred by independent binding of Rere to Wdr5 and to Hdac1/Hdac2.
- The delayed elution of Hdac1/Hdac2 shown in Fig. 1f does not correlate with the presence of Wdr5. This is in good correlation with the results of our co-immunoprecipitation experiments, showing no direct interaction between Wdr5 and Hdac1/Hdac2. This delayed elution of Hdac1/Hdac2 could be explained by the existence of multiple Hdac1/Hdac2-containing complexes other than WHHERE, such as the Sin3, NuRD and CoREST complexes.

The ChIP-assays of histone mark association with RARb should be included in the main manuscript.

- As suggested by the reviewer, the ChIP assays with the different histone marks have been moved to Fig. 6.

The reporter assay in NIH3T3 cells should include a construct with a mutated RARE. This would address if the signaling is a generic reporter artefact, or indeed through the RARE.

- The RARE-Luciferase reporter used in this study for cell cultures experiments uses the same promoter as the RA reporter used in the mouse line RARE-LacZ (Rossant et al. Genes Dev. 1991) (also used in this study) that has been widely used in RA research projects. This reporter is highly sensitive for RA signalling, different groups have extensively characterized it and it's a standard RA reporter used in cell and developmental biology papers (Blanco et al. Genes Dev. 1998; Lin et al. Nature 1998; Hoffman et al. J Cell Biol. 2006; Vilhais-Neto et al. Nature 2010; Cunningham et al. Genesis 2011; Papi et al. Cell Death Differ. 2012). Due to the high specificity of the reporter for RA signalling, comparison to the mutated form is not necessary and has not been performed in the studies cited above.

The Hdac1/2 association with the RAR β promoter was assessed after only 1 hr of RA treatment (Figure 4), which is before transcript levels increase. If high Hdac1/2 levels are maintained at later time points this could indeed support the controversial function of Hdac1/2 as a coactivator.

- After 1 hour of RA treatment, we observe an increase in Hdac1/Hdac2 levels at the *Rar β* promoter (Fig. 4h) together with Pol II (Fig. 4i). This suggests that Hdac1/Hdac2 (or WHHERE complex) could be important for the loading of the Pre-Initiation complex and Pol II for the initiation of transcription and early activation of RA signalling. If HDACs were acting as corepressors, we would expect them to be displaced from the *Rar β* promoter after 1 hour of RA treatment. We also show that knockdown of *Hdac1* or TSA treatment decreased Pol II recruitment (Fig. 4j,k) affecting RA signalling activation after 1 hour (Fig. 4a,b).

References: Overall appropriate references are cited, but are on occasions presented in a way which overstates agreement with and/or oversells the novelty of the current findings. As an example, a complex of Rere/Hdac1/Hdac2/Ehmt2 was already identified in *Drosophila* (as is correctly noted), but was reported to function by repressing transcription (Wang L, Charroux B, Kerridge S, Tsai CC. EMBO Rep. 2008 Jun;9(6):555-62). It is puzzling why this complex

would be repressing in *Drosophila*, while activating in mouse - particularly since most of the components as stated in the manuscript have known repressive functions.

- In the paper cited above, the authors describe binding of Hdac1, Hdac2 and Ehmt2 to Rere. They do not demonstrate the formation of a protein complex. In our work, we demonstrate using gel filtration experiments that Rere forms a stable protein complex (WHHERE) with Hdac1/Hdac2 and also with a newly identified partner (Wdr5) that can function as a positive regulator of RA signalling to control somite bilateral symmetry. While we show that Ehmt2 is also acting together with WHHERE in the positive regulation of RA signalling, it is likely to exhibit a very different, much more transient type of interaction with Rere, as only low NSAF values are observed in the Rere immunoprecipitations when compared to Hdac1/Hdac2 and Wdr5.

Reviewer #3 (Remarks to the Author):

Vilhais-Neto et al., have undertaken a proteomic approach to discover a coactivator complex that regulates retinoic acid (RA) signalling. Core epigenetic regulators (histone deacetylases and a histone methyl transferase) has been discovered to be part of this coactivator complex. They have followed this with a biochemical and cell-based analysis of its effects on RA-mediated transcription. This characterization appears to be detailed and thorough.

Mutations that target some of the new constituents were observed to disrupt left-right symmetries during somitogenesis in mouse embryos. However, there is no information about whether this complex is active in other embryonic tissues or whether it is a feature only of the paraxial mesoderm. Thus, even though the authors propose a link between epigenetic regulators and embryonic symmetry, it is unclear to what extent the effects of the mutations are specific towards embryonic symmetry. Overall, the paper does not yield any new insights into the mechanisms by which left-right symmetry is established.

- So far only two genes, *Raldh2* and *Rere*, both acting in the retinoic acid pathway, have been shown to regulate somite bilateral symmetry in the mouse. In this work we characterized three new players (Wdr5, Hdac1 and Ehmt2) involved in a coactivator complex (WHHERE) acting in the RA-dependent control of somite symmetry. Surprisingly, these new components have not been previously reported as being involved in RA signalling activation. While much remains to be done, we believe that our work significantly extends our understanding of the molecular regulation of bilateral symmetry downstream of retinoic acid signalling. Whether this Rere-dependent signalling pathway controls other aspect of embryo symmetry is certainly an interesting question but it is beyond the scope of this study.

The following comments need to be addressed:

1) Histone deacetylases and methyl transferases are ubiquitous long-lived epigenetic proteins involved with many different transcription factors. And, as the authors have mentioned, Rere is already known to interact with hdac1 and hdac2. Is it therefore an expected result that a third epigenetic regulator (Ehmt2) is also a part of the complex? How does this significantly increase our understanding of RA control of transcription?

- We do not claim that Ehmt2 is part of the complex but that it interacts with Rere to control the function of the WHHERE complex in RA signalling. None of these components except

for *Rere* have been implicated in RA signalling thus far despite a very significant amount of work in this field (largely performed *in vitro*). Our data demonstrate the role of these proteins in controlling RA signalling *in vivo* and *in vitro* and show their functional role in a RA-dependent process, the control of bilateral symmetry. We believe this very significantly increases our understanding of RA control of transcription.

2) Both Fig. 2c and 2d display activation upon *Hdac2* siRNA, but they seem to have different levels of fold activation. Can the authors explain or clarify this?

- Fig. 2c corresponds to a single siRNA experiment: the concentration during transfection for each of the siRNAs (for *Rere*, *Wdr5*, *Hdac1* and *Hdac2*) was 100 nM. In the case of Fig. 2d it corresponds to double siRNA transfection. To maintain similar concentrations in all conditions during double siRNA assays the transfection used was 50 nM of siRNA for each target to make a total of 100 nM:

- Ctrl: 100 nM siRNA Ctrl
- *Hdac1*: 50 nM siRNA *Hdac1* + 50 nM siRNA Ctrl
- *Hdac2*: 50 nM siRNA *Hdac2* + 50 nM siRNA Ctrl
- *Hdac1/Hdac2*: 50 nM siRNA *Hdac1* + 50 nM siRNA *Hdac2*

The knockdown for *Hdac2* should be more efficient in Fig. 2c (100 nM of *Hdac2* siRNA) explaining the higher fold activation in this condition than in Fig. 2d (50 nM of *Hdac2* siRNA).

The siRNA amounts used during single or double siRNA transfections have been added to the methods in the revised version of the paper.

3) The authors observe a dramatic phenotype upon crossing *Rere* mutants with *Wdr5*-TCre, where 35% of the embryos lacked segmented structures on the right side. How do these mutations lead to much severe phenotypes than RA synthesizing mutants? Does this suggest that *Wdr5* and *Rere* are likely to be involved in other signalling pathways regulating embryonic symmetry?

- A similar phenotype was observed in the *Rere*^{+/*om*}*Raldh2*^{-/-} mutant in our previous paper (Vilhais-Neto et al. Nature 2010). At present we cannot explain such an interesting asymmetric phenotype but one of the hypothesis could be that different (other than RA) signalling pathways could be affected in the double mutants. This data clearly demonstrates genetic interaction between *Wdr5* and *Rere* indicating that they act in the same pathway.

4) It is unclear how any arguments on symmetry versus asymmetry can be made from the phenotype of *Wdr5* mutants (Fig. 3h), which have an overall altered embryonic morphology and highly disrupted somitogenesis. From the data presented, one can conclude that somitogenesis is disrupted, but one cannot conclude that this is a specific phenotype, i.e. that somites are affected independently of any other structure, nor can one conclude that symmetry is affected.

- *Wdr5* mutant embryos have somite desynchronization defects in a limited number of embryos (Fig. 3h,m). Intercrossing *Wdr5* and *Rere* mutants clearly demonstrates interplay between these two molecules on the regulation of embryonic somite symmetry showed by the lack of right-segmented side (Fig. 3o). This was observed in several embryos and indicates a major alteration of the bilateral symmetry of somitogenesis.

5) Can the authors comment on why Rar α was not found to be increased on the Rar β promoter in the ChIP experiments (Fig. 4g,h) performed after 1 hr of RA treatment?

- We and others have demonstrated that Rar α bound at the RARE element on the *Rar β* promoter does not increase after RA treatment (Pavri et al. Mol Cell. 2005; Gillespie and Gudas J Mol Biol. 2007; Vilhais-Neto et al. Nature 2010). One would expect an increase in the coactivators like presented in Fig. 4h but not for Rar α .

6) Based on the observed pull-down of Hdac1 and Rar α (Fig. 4l-n), the authors suggest that Hdac1 might connect the WHHERE complex to Rar α , which seems a pretty strong statement. The data show that the proteins can be found in in the same complex, and nothing more.

- We agree with the reviewer comment and have rephrased the sentence.

Reviewer #4 (Remarks to the Author):

The authors of the manuscript entitled: "A new coactivator complex required for retinoic acid-dependent regulation of embryonic symmetry" describe the isolation of a set of proteins which are required for somitogenesis symmetry in mice. They also bring evidence that these proteins bind to the RARE element and induce transcription of a RARE driven reporter gene in the cell and in vivo. Amongst those proteins is WDR5, which is a novel factor involved in RA-driven vertebrate bilateral symmetry. They also present evidence that inhibition of HDACs interferes with bilateral symmetry and that the protein methyltransferase EHMT2 has a positive role in symmetric somite formation. Albeit the in vivo and the functional data presented by the authors are compelling, several aspects of this work require major revision prior publication in this journal.

1) WHHERE complex:

a) Although the identification of the Rere interaction partners in the transgenic mouse is indisputable, the novelty of the WDR5 binding to Rere (line 75) has to be mitigated since the interaction between WDR5 and Rere has been already published in Hein et al., Cell 2015 (reference to be included).

- Hein et al. identified Rere binding to Wdr5 in a paper analyzing the entire human interactome (A human interactome in three quantitative dimensions organized by stoichiometries and abundances. Cell. 2015 Oct 22;163(3):712-23). Even though this does not provide evidence for direct binding between Rere and Wdr5, nor for participation to a stable protein complex, we included the reference per the reviewer suggestion. In the revised manuscript we provide evidence for direct binding between Rere and Wdr5 (Fig. 1d).

b) Also the claim for the direct interaction between WDR5 and Rere (Fig 1d) requires the proper control which is an IP with anti -Flag antibodies from extract containing only HA-WDR5. Moreover, the authors should exclude the presence of endogenous HDAC1/2 in those IPs (Fig 1d).

- We have included new results showing no direct interaction between Wdr5 and Hdac1/Hdac2 (Supplementary Fig. 2c) from co-immunoprecipitation experiments performed from baculovirus-insect cells co-expressing HA-Wdr5 and Flag-Hdac1/Flag-Hdac2. This

demonstrates that Wdr5 only binds Rere and not Hdac1/Hdac2. No cross interaction could be detected between Flag and HA antibodies on Supplementary Fig. 2c.

c) For the protein association data after co-expression in the baculovirus system (Fig. 1c, 1e and 1f), the author should explain and show the specificity of the M2 -resin pull-down for either HDAC1, HDAC2 or Rere which are all tagged with Flag before concluding on complex formation

- The interaction Rere/Hdac1/Hdac2 was previously reported in: Zoltewicz et al. Development 2004; Wang et al. Genes Dev. 2006; Wang et al. EMBO Rep. 2008 and our study extends this earlier finding by adding Wdr5 to the complex. The goal of the experiment described in Fig. 1c was to validate the identification of Wdr5, as a novel binding partner of Rere/Hdac1/Hdac2. By performing Flag immunoprecipitation of Flag-tagged Rere, Hdac1 and Hdac2, we ensured to pull down the three interacting proteins while validating the Wdr5 interaction by using a different tag, HA-tag. The existence of the WHHERE complex was demonstrated by gel filtration experiment described in Fig. 1e,f. We have also clarified the conclusions in the text.

- In the revised version, we also include results showing no direct interaction between Wdr5 and Hdac1/Hdac2 (Supplementary Fig. 2c) from co-immunoprecipitation experiments performed from baculovirus-insect cells co-expressing HA-Wdr5 and Flag-Hdac1/Flag-Hdac2. This would suggest that Rere binds independently to Wdr5 and to Hdac1/Hdac2.

a) Stoichiometry of the WHHERE complex components: the authors use spectral counts (NSAF) to calculate stoichiometry. Since spectra counting suffers from a poor precision, there are more advanced methods nowadays to estimate complex composition which I would encourage to apply as the one in published in Hein et al , Cell 163, Issue 3, 2015. For a precise determination of a the stoichiometry methods e.g. AQUA peptides, SILAC or iTRAQ/TMT are available.

- We agree with this referee that precise determination of stoichiometry would require different methods but we believe the study of the complex stoichiometry is beyond the scope of this paper. We have removed the statement about stoichiometry from the text.

2) Functional assay:

a) The cellular reporter assay shown in Fig 2 albeit reproducible within the same experiment (very small error bars) suffers from a certain variability between different experiments (i.e. compare Figs. 2a and 2b after HDAC1 overexpression, Figs 2d and 2f for the effect of HDAC1 siRNA or HDAC1 and HDAC2 si RNA after WDR5 overexpression). It is therefore hard to conclude on the effect of the double knock-down of the HDACs if the experiments are not done side by side with the single knock-downs. Alternatively, the statement in line 111 should be removed.

- For single overexpression experiments (Fig. 2a) the amount of each construct (for Rere, Wdr5, Hdac1 and Hdac2) transfected was 200 ng. In co-transfection experiments (Fig. 2b) the amount transfected for each of the plasmids was 150 ng to make a total of 300 ng:

- Ctrl: 300 ng pFlag
- Hdac1: 150 ng Hdac1 + 150 ng pFlag
- Hdac2: 150 ng Hdac2 + 150 ng pFlag
- Hdac1/Hdac2: 150 ng Hdac1 + 150 ng Hdac2

Since a higher amount of plasmid was used for Hdac1 construct in Fig. 2a (200 ng) we should expect stronger activation of RA signalling than in Fig. 2b (150 ng).

- In single siRNA experiments (Fig. 2c,e): the concentration during transfection for each of the siRNAs (for *Rere*, *Wdr5*, *Hdac1* and *Hdac2*) was 100 nM. In the case of Fig. 2d,f it corresponds to double siRNA transfection. To maintain similar concentrations in all conditions during double siRNA assays the transfection used was 50 nM of siRNA for each target to make a total of 100 nM:

- Ctrl: 100 nM siRNA Ctrl
- Hdac1: 50 nM siRNA Hdac1 + 50 nM siRNA Ctrl
- Hdac2: 50 nM siRNA Hdac2 + 50 nM siRNA Ctrl
- Hdac1/Hdac2: 50 nM siRNA Hdac1 + 50 nM siRNA Hdac2

The plasmid and siRNA amounts used during single or double transfections have been added to the methods in the revised version of the paper.

b) To validate the unexpected transcriptional activation effect of the knock down of HDAC2 (Fig 2c) the authors should show the effect of the HDAC2^{-/-} mice mutant on the expression of the RARE LacZ reporter.

- As reported by others and described in the manuscript, *Hdac2* mutants do not show developmental defects and can survive until the perinatal period (Montgomery et al. Genes Dev. 2007). As the *Hdac2* KO has no phenotype, while interesting, we feel this study is beyond the scope of this paper.

c) In order to clearly demonstrate an RA-dependent activation role of the WHHERE complex the authors should show reporter data in the absence of RA treatment following overexpression or knock-down of the different proteins (and not just of control).

- The promoter of the RARE-Luciferase reporter used in this study for cell cultures experiments is the same as the one used in the mouse line RARE-LacZ (Rossant et al. Genes Dev. 1991) (also used in this study) which has been widely used in RA studies by a wide variety of investigators. This reporter is highly sensitive for RA signalling, and has been extensively characterized by many groups and used as a standard RA reporter in many cell and developmental biology papers (Blanco et al. Genes Dev. 1998; Lin et al. Nature 1998; Hoffman et al. J Cell Biol. 2006; Vilhais-Neto et al. Nature 2010; Cunningham et al. Genesis 2011; Papi et al. Cell Death Differ. 2012). In a previous paper (Vilhais-Neto et al., Nature 2010) we have performed the same overexpression and knockdown experiments with *Rere* with and without RA and we concluded that this reporter is only active after RA treatment.

Moreover, they should show the effect of at least 1 example of knock down or overexpression of a known RA activator (i.e. EP300). A combination of the latter with any of the WHHERE complex components, might also give insight into the molecular mechanism of HDAC-mediated increase of expression of RA-dependent genes (indeed HDACs have been already shown to increase gene expression of nuclear receptor controlled genes by acting, amongst others, on PolII recruitment _Welsbie et al 2009)

- Ep300 has been shown to acetylate Hdac1 and inactivates its deacetylase activity (Qiu et al. Mol Cell. 2006). In agreement with these findings, we now show in the revised version that Ep300 inhibits RA signalling and reduces Hdac1 activation of RA signalling (Fig. 7a). Furthermore, we show that a mutant form of Hdac1 (H1-6R) resistant to Ep300 acetylation

activates RA signalling more strongly than the Hdac1 wild-type form (H1-WT) (Fig. 7b). We also report that Ep300 inhibits Rara- and Rere-dependent RA signalling whereas Kat2a (Gcn5) has a positive role on Rara- and Rere-dependent RA activation (Fig. 7c,d). While siRNA mediated knockdown of Rere or Kat2a in the presence of RA decreased RA signalling, knockdown of Ep300 did not affect the RA pathway (Fig. 7e). Similarly in Ep300 mutant embryos (*Ep300^{-/-}*) RARE-LacZ reporter expression appeared normal (Fig. 7f,g) and somitogenesis progressed symmetrically (Fig. 7h,i). These results demonstrated that Ep300 negatively regulates Hdac1-dependent activation of RA signalling, and Kat2a can participate together with the WHHERE complex in the activation of the RA pathway.

3) Interaction with EHMT2:

a) The weak retrieval of the EHMT2 protein in the complex might also be due to substoichiometric interaction (to consider in line 187).

In any case, the statement in line 206 on a scaffolding role for EHMT2 for RA regulated genes, should be mitigated since the retrieval of EHMT2 with the WHHERE complex is weak (Fig 5a), and EHMT2 binds to the RarB promoter also in absence of RA (fig 5c).

We are not claiming that Ehmt2 is part of the WHHERE complex but rather that it interacts with Rere to control the complex function in RA signalling. Ehmt2 has been shown to bind the N-terminal region of Rere (Wang et al. EMBO Rep. 2008). Our work shows that:

- The knockdown of *Ehmt2* decreases WHHERE complex on the *Rarβ* promoter (Fig. 5k) leading to a decrease of WHHERE-dependent RA signalling (Fig. 5i).

- Even though Ehmt2 can be found on the *Rarβ* promoter it increases rapidly after 1 hour of RA treatment (Fig. 5j) like the WHHERE complex components (Fig. 4h).

- Finally *Ehmt2* embryo mutants have clear reduced RA signalling (Fig. 5a,b) and asymmetric somitogenesis as observed for most WHHERE members (Fig. 5c,d). This is consistent with the results of cell culture assays showing reduced RA signalling after *Ehmt2* siRNA-mediated knockdown (Fig. 5f).

This is also in agreement with no change in the level of H3K9me1 or of H3K9me2 at this promoter. The use of potent and selective G9a inhibitor might also clarify whether the methyltransferase activity of EHMT2 is required for this function (Kubicek S et al. (2007) Reversal of H3K9me2 by a small-molecule inhibitor for the G9a histone methyltransferase. Mol Cell 25: 473-481)

- For the inhibition of Ehmt2 methyltransferase activity we used two potent and selective inhibitors: UNC0638 (U38) and UNC0646 (U46) (Vedadi et al. Nat.Chem.Biol. 2011; Liu et al. J.Med.Chem. 2011). In the revised version, we show that inhibition of Ehmt2 methyltransferase activity with U38 or U46 *in vitro* does not alter RA signalling in treated cells (Supplementary Fig. 5e,f). Also no difference in H3K9me1 nor H3K9me2 levels was observed by ChIP after 1 hour of RA treatment suggesting that Ehmt2 function is likely to be independent from its methyltransferase activity (Supplementary Fig. 5g,h).

b) The authors should indicate the position of EHMT2 in Suppl fig 1f and/or 1e

For the clustering analysis shown in Supplementary Fig. 1e and f, we only used the 105 common proteins found between the four different immunopurification conditions (HS-S: High Salt and Sigma HA beads, LS-R: Low Salt and Roche HA beads, LS-S: Low Salt and Sigma HA beads, HS-R: High Salt and Roche HA beads). Since Ehmt2 (and Ehmt1) was not found in all four conditions it was not included in the hierarchical clustering analysis.

Reviewers' Comments:

Reviewer #2 (Remarks to the Author)

Response to revisions:

In the current manuscript the authors confirm interactions between Rere, Hdac1, and Hdac2, which have been previously demonstrated, and adds Wdr5 to the complex. In vitro reporter activity mirrors Hdac1 levels, whereas either increased or decreased Hdac2 levels both result in elevated reporter activity (Fig.2). The reporter contains an RA-responsive element, which as previously demonstrated, binds RAR/RXR hetero-dimers to mediate transcriptional activation. The question remains whether the WHHERE complex works through this element, or through other cis-acting element. Inclusion of a mutated reporter construct is required in order to determine whether reporter activity in general is elevated or whether it is (as claimed) the RA-response of the reporter that is elevated. In other words, because the reporter may respond to conditions unrelated to RA, an evaluation of the RA-responsive element needs to be included.

The authors point out that much of the work, which has established Hdacs as repressors, is based on tissue culture assays, which "relevance in vivo remains to be established". Aside the fact that the authors themselves base most of their functional on tissue culture assays, the WHHERE complex was actually demonstrated in vivo (*Drosophila*) to have repressive functions. An alternative model could propose that the functions of certain nuclear receptors are inhibited by acetylation, and that if receptor acetylation is favored, the transcription is dampened.

Overall, the amount of data is very impressive, yet it appears have come about at the expense of controls, which if they had been included would have facilitated more conclusive interpretations.

Specific comments:

The delayed elution of Hdac1/2 relative to Rere/Wdr5 (Fig.1f) suggests that only a small fraction of the Hdac1/2 is in complex with Rere/Wdr5, yet the revised figure (Fig.1g) seems to indicate Rere:Hdac1:Hdac2:Wdr5 ratios in the Rere-IP similar to those in the input.

Overexpression of N-Rere, but not C-Rere, is shown to elevate reporter activity (Fig.2j). Because this is evaluated neither on an RARE-deficient reporter nor in control conditions (absence of RA) a clear distinction cannot be made between the RA-response of the reporter and general reporter activity (which can in some instances be elevated by inhibition of p300/CBP-mediated acetylation).

Overexpression of p300 had a negative effect on reporter activity (Fig.7a). This not surprising, yet still does not address why acetylation negatively regulates reporter activity. Over expression of an acetylation resistant Hdac1 (6R) suggests that Hdac1 may be the target (Fig.7b). Extending on the reporter assays, the authors add Kat2a to the list of positive regulators (Fig.7c). This is interesting because although p300 and Kat2a both catalyze protein acetylation, the targets differ between the two. It would be interesting to see if an acetylation resistant RARalpha would also lead to a more potent reporter activation, but the reporter assays shed little light on the biological function (is RARbeta activation even a relevant RA target when considering bilateral symmetry).

It is difficult to determine if the CoIP protocol was successful (Sup.Fig.2c) since no positive control (e.g. Rere) was included. As such, the absence of evidence cannot be taken as evidence of absence (ruling out Wdr5/Hdac1/2 interactions).

Reviewer #3 (Remarks to the Author)

In general I think the revised manuscript is improved.

I have two remaining concerns that regard the way the embryological data is presented and interpreted.

Symmetry and synchronization.

When the authors write that a mutation has affected somite synchronization, do they mean a

change to the left-right symmetry or the synchronization of individual oscillating cells in the tissue? In some cases it appears that these are used as synonyms, but in others the observed somite disruption is strong and potentially consistent with a desynchronization of oscillators. Since the terms synchronization and synchrony have a specific meaning in dealing with oscillators (i.e. active and mutual entrainment Pikovsky, Rosenblum and Kurths, 2001), and as currently there is no evidence to suggest that this is the mechanism ensuring left-right symmetry, I urge the authors to distinguish between the concepts by careful choice of the terms.

Specificity of symmetry defects.

From the response of the authors to my query regarding the specificity of symmetry defects, I fear that I may not have made my concerns clear, so I will try another tack in explanation.

In the introduction, the authors introduce the concept of left-right symmetry in somitogenesis at the end of page two. "In the absence of RA signalling in the mouse embryo, somite formation becomes asymmetrical, showing a significant delay on the right side 7. A similar somite desynchronization phenotype is also observed in mutants for the protein Rere (or Atrophia2) which acts as a coactivator for RA signaling 8."

From this description, the authors appear to have defined somite asymmetry resulting from a loss of RA signaling as a delay, presumably defined by somites on the right side forming later than somites on the left hand side. However, in the embryos in Figure 3 etc., a range of defects to the somites is displayed, ranging from a right-handed delay, to defects in somite boundary shape or alignment at various locations along the axis (with or without delay at the posterior), to a rather profound disruption of segmentation where it is challenging to interpret a miss-match of left and right-hand sides. In the most severe cases, even the size and shape of the embryo appear to be altered. However, in the text, and in the quantification of defects (figure 3ijk) these effects appear to be combined or pooled together as if they are the same thing, for example: "... (i-k) Graphs representing the number of 7- to 15-somite stage embryos with left-sided (orange), symmetric (blue) or right-sided (green) delay in somite formation:..." I am not sure what the common practice in the field is when describing such defects, but I do not agree that this is an accurate description of the embryos. Consequently, statements such as "Mouse mutants for Wdr5 and Hdac1 exhibit asymmetrical somite formation characteristic of RA deficiency." in the abstract appear to be oversimplifications.

Thus, although it is obvious that many of the embryos have left-right differences, it is not clear to me that somite symmetry is specifically or even primarily affected. In my opinion, the authors need to be much more precise about these issues by (a) defining carefully what constitutes a somite asymmetry, and being consistent with this, (b) acknowledging the heterogeneity of phenotypes and the range of severity and how these correlate with the different gene knockouts (I think this is fascinating, by the way, because it may hint at overlapping or branching effects in the pathways).

Reviewer #4 (Remarks to the Author)

The authors have addressed all criticisms raised satisfactorily.

I now recommend the publication of the manuscript in Nature Communications

Response to the reviewers

Reviewer #2 (Remarks to the Author):

Response to revisions:

In the current manuscript the authors confirm interactions between Rere, Hdac1, and Hdac2, which have been previously demonstrated, and adds Wdr5 to the complex.

- While this is true, these interactions were demonstrated in the context of a co-repressor activity while in this paper we demonstrate coactivator properties of the Rere complex containing Hdac1 and Hdac2. Formation of a stable complex was not established in these papers.

In vitro reporter activity mirrors Hdac1 levels, whereas either increased or decreased Hdac2 levels both results in elevated reporter activity (Fig.2). The reporter contains an RA-responsive element, which as previously demonstrated, binds RAR/RXR hetero-dimers to mediate transcriptional activation. The question remains whether the WHHERE complex works through this element, or through other cis-acting element. Inclusion of a mutated reporter construct is required in order to determine whether reporter activity in general is elevated or whether it is (as claimed) the RA-response of the reporter that is elevated. In other words, because the reporter may respond to conditions unrelated to RA, an evaluation of the RA-responsive element needs to be included.

- The RA specificity of the reporter used in our work has been addressed in several publications including our last report in Nature (Vilhais-Neto et al, Nature 2010). Nevertheless, to definitively address the reviewer's concerns on the specificity of the RARE-Luciferase reporter and of the WHHERE-dependent RA activation, we performed experiments in cell culture in the absence and presence of RA after overexpression of each of the WHHERE complex components (Rere, Wdr5, Hdac1 and Hdac2) together with the reporter. We show that overexpression of Rere, Wdr5, Hdac1 and Hdac2 activate the RARE-Luciferase reporter only in the presence of RA. This data is now shown in Supplementary Fig. 3a. We also used two different reporters, which do not respond to RA: pGL3-Promoter-Luciferase (luciferase driven by the SV40 promoter) and TA-Luciferase (minimal TATA box promoter driving luciferase). No activation of pGL3-Promoter-Luciferase or TA-Luciferase was observed after overexpression of Rere, Wdr5, Hdac1 and Hdac2 either in the absence or presence of RA. This new data is now shown in Supplementary Fig. 3b,c.

The authors point out that much of the work, which has established Hdacs as repressors, is based on tissue culture assays, which “relevance in vivo remains to be established”. Aside the fact that the authors themselves base most of their functional on tissue culture assays, the WHHERE complex was actually demonstrated in vivo (Drosophila) to have repressive functions. An alternative model could propose that the functions of certain nuclear receptors are inhibited by acetylation, and that if receptor acetylation is favored, the transcription is dampened.

- We thank the referee for this interesting suggestion. The regulation of acetylation of nuclear receptors is an interesting hypothesis to explain the control of RA activation in our context. It is consistent with our observation that Ep300 represses RA signalling, which could be caused by acetylation of the RARs. In turn, repression could be relieved

by deacetylation of the RARs mediated by Hdac1 and Hdac2, which would promote RA signalling. We have introduced this idea in the discussion of the revised text.

Overall, the amount of data is very impressive, yet it appears have come about at the expense of controls, which if they had been included would have facilitated more conclusive interpretations.

- In the new revised version we have now demonstrated the specificity of the RA activation by the WHHERE complex as explained above (Supplementary Fig. 3a-c).

Specific comments:

The delayed elution of Hdac1/2 relative to Rere/Wdr5 (Fig.1f) suggests that only a small fraction of the Hdac1/2 is in complex with Rere/Wdr5, yet the revised figure (Fig.1g) seems to indicate Rere:Hdac1:Hdac2:Wdr5 ratios in the Rere-IP similar to those in the input.

- In Figure 1f, the peak of Hdac1 and Hdac2 is slightly offset compared to the peak of Rere and Wdr5, suggesting the existence of lower molecular weight complexes involving Hdac1 and Hdac2 but not Rere and Wdr5. Hdac1 and 2 are well known to form homo and heterodimers, which could elute in this peak. Nevertheless, significant amounts of Hdac1 and Hdac2 are observed in the fractions containing the peak of Rere and Wdr5 (A10-B1) consistent with the existence of the WHHERE complex.

- Figure 1f shows the result of a gel filtration analysis with purified WHHERE components produced in baculovirus while Figure 1g shows the result of IPs of NIH3T3 cell extracts using the Rere antibody. We do not think it is possible to make any quantitative claims comparing these two experiments performed in very different conditions (in vitro and in vivo).

Overexpression of N-Rere, but not C-Rere, is shown to elevate reporter activity (Fig.2j). Because this is evaluated neither on an RARE-deficient reporter nor in control conditions (absence of RA) a clear distinction cannot be made between the RA-response of the reporter and general reporter activity (which can in some instances be elevated by inhibition of p300/CBP-mediated acetylation).

- In the new revised version we demonstrate the specificity of the RA activation by the WHHERE complex as discussed above. We performed experiments in absence and presence of RA using the RARE-Luciferase, pGL3-Promoter-Luciferase or TA-Luciferase. The WHHERE complex only activates RARE-Luciferase in the presence of RA (Supplementary Fig. 3a-c). Interestingly inhibition of Ep300 through siRNA did not alter the RARE-Luciferase response (Fig. 7e). In contrast, overexpression of Ep300 inhibited RA signalling (Fig. 7a,c,d).

Overexpression of p300 had a negative effect on reporter activity (Fig.7a). This not surprising, yet still does not address why acetylation negatively regulates reporter activity. Over expression of an acetylation resistant Hdac1 (6R) suggests that Hdac1 may be the target (Fig.7b). Extending on the reporter assays, the authors add Kat2a to the list of positive regulators (Fig.7c). This is interesting because although p300 and Kat2a both catalyze protein acetylation, the targets differ between the two. It would be interesting to see if an acetylation resistant RARalpha would also lead to a more potent reporter activation, but the reporter

assays shed little light on the biological function (is RARbeta activation even a relevant RA target when considering bilateral symmetry).

- We agree with the reviewer that overexpression of an acetylation resistant form of Rara would be an interesting experiment, however we feel this work is beyond the scope of this study.

- In mouse embryos, Rarb mRNA exhibits a similar expression pattern to RARE-LacZ and it is severely downregulated in the paraxial mesoderm of the Ralhd2 null mouse mutants (Niederreither et al. Development 2002) indicating that it is a relevant RA target. Also, in cell culture, Rarb mRNA and RARE-Luciferase have similar kinetics after adding RA to the media (Fig. 4a,b).

It is difficult to determine if the CoIP protocol was successful (Sup.Fig.2c) since no positive control (e.g. Rere) was included. As such, the absence of evidence cannot be taken as evidence of absence (ruling out Wdr5/Hdac1/2 interactions).

- We have now included in Supplementary Fig. 2c an IP containing Rere, Hdac1, Hdac2 and Wdr5 in high salt conditions. In these conditions, HA-Wdr5 pull-down from cells infected with baculoviruses expressing HA-Wdr5, Rere-Flag, Flag-Hdac1 and Flag-Hdac2 contained the four proteins, which supports the notion that Rere acts as a scaffold component for the WHHERE complex.

Reviewer #3 (Remarks to the Author):

In general I think the revised manuscript is improved.

I have two remaining concerns that regard the way the embryological data is presented and interpreted.

Symmetry and synchronization.

When the authors write that a mutation has affected somite synchronization, do they mean a change to the left-right symmetry or the synchronization of individual oscillating cells in the tissue? In some cases it appears that these are used as synonyms, but in others the observed somite disruption is strong and potentially consistent with a desynchronization of oscillators. Since the terms synchronization and synchrony have a specific meaning in dealing with oscillators (i.e. active and mutual entrainment Pikovsky, Rosenblum and Kurths, 2001), and as currently there is no evidence to suggest that this is the mechanism ensuring left-right symmetry, I urge the authors to distinguish between the concepts by careful choice of the terms.

- Oscillations of the segmentation clock terminate before somite formation thus we do not see how one could confuse somite synchronization with the synchronization of segmentation clock oscillations. In the entire manuscript, the somite synchronization defects refer to changes in left-right somite symmetry resulting from desynchronization of the somite formation process associated to a delay in somite formation on one side.

Specificity of symmetry defects.

From the response of the authors to my query regarding the specificity of symmetry defects, I fear that I may not have made my concerns clear, so I will try another tack in explanation.

In the introduction, the authors introduce the concept of left-right symmetry in somitogenesis at the end of page two. “In the absence of RA signalling in the mouse embryo, somite formation becomes asymmetrical, showing a significant delay on the right side 7. A similar somite desynchronization phenotype is also observed in mutants for the protein Rere (or Atrophia2) which acts as a coactivator for RA signaling 8.”

From this description, the authors appear to have defined somite asymmetry resulting from a loss of RA signaling as a delay, presumably defined by somites on the right side forming later than somites on the left hand side. However, in the embryos in Figure 3 etc., a range of defects to the somites is displayed, ranging from a right-handed delay, to defects in somite boundary shape or alignment at various locations along the axis (with or without delay at the posterior), to a rather profound disruption of segmentation where it is challenging to interpret a miss-match of left and right-hand sides. In the most severe cases, even the size and shape of the embryo appear to be altered. However, in the text, and in the quantification of defects (figure 3ijk) these effects appear to be combined or pooled together as if they are the same thing, for example: “... (i-k) Graphs representing the number of 7- to 15-somite stage embryos with left-sided (orange), symmetric (blue) or right-sided (green) delay in somite formation:...”. I am not sure what the common practice in the field is when describing such defects, but I do not agree that this is an accurate description of the embryos. Consequently, statements such as “Mouse mutants for *Wdr5* and *Hdac1* exhibit asymmetrical somite formation characteristic of RA deficiency.” in the abstract appear to be oversimplifications.

Thus, although it is obvious that many of the embryos have left-right differences, it is not clear to me that somite symmetry is specifically or even primarily affected. In my opinion, the authors need to be much more precise about these issues by (a) defining carefully what constitutes a somite asymmetry, and being consistent with this, (b) acknowledging the heterogeneity of phenotypes and the range of severity and how these correlate with the different gene knockouts (I think this is fascinating, by the way, because it may hint at overlapping or branching effects in the pathways).

- We respectfully disagree with this referee. We are careful to say in the text that we score embryos for the presence of a right side delay in somite formation. This is very precise and characteristic of RA deficiency as reported in Vermot et al. Science 2005 or in Vilhais-Neto et al. Nature 2010. It is true that in some of the mutants such as *Wdr5*, somites also exhibit abnormal shapes in addition to the delay phenotype. This is not unexpected given the pleiotropic role of *Wdr5* in the regulation of transcription. We have clarified this point in the text of the revised version.

Reviewers' Comments:

Reviewer #2:

Remarks to the Author:

Additional controls were included in the revised manuscript. This increases the confidence in the conclusions drawn from the experimental findings. It remains puzzling how a complex of co-repressors can activate transcription (particular since in *Drosophila* a similar complex functions as a transcriptional repressor), but unfortunately this conundrum will not be resolved in the presented manuscript.

Apart from that, the points of my critique were addressed.

The manuscript is acceptable for publication in *Nature Communications*.